# High-resolution discrimination of homologous and isomeric proteinogenic amino acids in nanopore sensors with ultrashort single-walled carbon nanotubes

Weichao Peng[1,2], Shuaihu Yan[2,3], Ke Zhou[3], Hai-Chen Wu ®[2,3] ✉, Lei Liu ®[1] ✉ & Yuliang Zhao ®[1,4]

The hollow and tubular structure of single-walled carbon nanotubes (SWCNTs) makes them ideal candidates for making nanopores. However, the heterogeneity of SWCNTs hinders the fabrication of robust and reproducible carbon-based nanopore sensors. Here we develop a modified density gradient ultra-centrifugation approach to separate ultrashort (≈5-10 nm) SWCNTs with a narrow conductance range and construct high-resolution nanopore sensors with those tubes inserted in lipid bilayers. By conducting ionic current recordings and fluorescent imaging of $Ca^{2+}$ flux through different nanopores, we prove that the ion mobilities in SWCNT nanopores are 3-5 times higher than the bulk mobility. Furthermore, we employ SWCNT nanopores to discriminate homologue or isomeric proteinogenic amino acids, which are challenging tasks for other nanopore sensors. These successes, coupled with the building of SWCNT nanopore arrays, may constitute a crucial part of the recently burgeoning protein sequencing technologies.

The hollow and tubular structure endows carbon nanotubes (CNTs) with many peculiar properties for a wide variety of applications such as molecular transport[1–4], stochastic sensing[5–7], transistor with endohedral doping[8], and catalysis in confinement[9,10]. As for molecular sensing with CNTs, the earlier works had been focused on the surface modifications of CNTs and the fabrication of field effect transistors[11–14]. In 2010, Liu et al. fabricated a device where a 2-µm-long SWCNT connected two fluid reservoirs, and reported the translocation of single-stranded DNA (ssDNA) through a SWCNT[5]. It was the beginning that the tubular structure of CNTs started to be explored for sensing applications. Soon after that, our group adopted a different strategy of acquiring a single SWCNT in a lipid bilayer and fabricated nanopore sensors based on ultrashort SWCNTs (5–10 nm)[6]. This construct was later reproduced by Geng et al. with a different procedure[7]. Both the

works reported the translocation of ssDNA through ultrashort SWCNTs as current blockades rather than current-increasing spikes observed by Liu et al.[5]. However, no more works on stochastic sensing with SWCNT nanopores have been reported ever since, partly due to lacking of a methodology for obtaining homogeneous SWCNTs.

Nanopore sensing relies on monitoring the ionic current fluctuation through nanopores when an analyte binds with the pore. The current amplitude changes and the mean dwell time of the analyte binding events reveal its identity, whereas the frequency of the binding events reveals the analyte concentration[15]. Generally, for the sensing of a specific target, one needs to install a recognizing element inside the nanopore, through chemical modification or site-directed mutagenesis in biological pores[16,17], or surface modification in solid-state nanopores[18,19]. However, SWCNT nanopores involves a different

[1]Key Laboratory for Biomedical Effects of Nanomaterials & Nanosafety, Institute of High Energy Physics, Chinese Academy of Sciences, Beijing 100049, China. [2]University of Chinese Academy of Sciences, Beijing 100049, China. [3]Beijing National Laboratory for Molecular Sciences, Key Laboratory of Analytical Chemistry for Living Biosystems, Institute of Chemistry, Chinese Academy of Sciences, Beijing 100190, China. [4]CAS Key Laboratory for Biomedical Effects of Nanomaterials & Nanosafety, National Center for Nanoscience and Technology, Beijing 100190, China. ✉e-mail: haichenwu@iccas.ac.cn; leiliu@ihep.ac.cn

underlying mechanism for molecular sensing. There are no either constrictions or any sensing elements inside SWCNT nanopores, but SWCNTs have an ideal tubular structure and atomically smooth surface. When a molecule traverses through a SWCNT nanopore under transmembrane potentials or by diffusion, it might interact with the inner wall of CNTs via non-covalent interactions such as π-π, CH-π, or hydrophobic interactions. Owing to the uniformity of the interior environment of CNTs, the current alterations caused by the molecule-CNT interactions may remain the same no matter where the interactions occur. This special feature would enhance sensing sensitivity for small molecules in SWCNT nanopores. In this report, we separate ultrashort SWCNTs (≈5–10 nm) with density gradient ultracentrifugation (DGU) and take appropriate fractions to construct nanopore sensors inserted in lipid bilayers (Fig. 1a). The SWCNT nanopores exhibit very high resolution in discriminating closely resembling molecules such as homologue and isomeric proteinogenic amino acids. We attribute this unusual capacity to the faster ion transport in SWCNT nanopores than in bulk solutions, which is corroborated by both the current recordings and optical imaging experiments. The high-resolution discrimination of similar targets such as amino acids in SWCNT nanopores will find useful applications in many scenarios, particularly in the recently boomed protein sequencing technologies.

## Results

### Fabrication and characterization of SWCNT nanopores

We obtained short SWCNTs through cutting long SWCNTs with sonication in concentrated sulphuric acid/nitric acid (3/1) and subsequent DGU separation (Supplementary Figs. 1–3 and Methods). Previously, our and Noy's group reported the synthesis of short SWCNTs with different methods[6,7], but both procedures showed limited control over the selection of nanotube diameters. In this work, we found that the DGU process has much better resolution in sorting SWCNT diameters[20], as we managed to record a quite narrow range of

conductance centered around 1.0 nS of 100 SWCNTs in a DGU fraction (Fig. 1b and Supplementary Figs. 4–6 for characterization)[21]. By assuming a very simple model without any free energy barrier, the conductance of a tube of electrolyte should be given by[5]

$$G = 6.02 \times 10^{26}(m_K + m_{Cl})c_{KCl}e\pi D^2(4L)^{-1} \quad (1)$$

where $\mu_K$ is the ionic mobility of K⁺, $\mu_{Cl}$ is the ionic mobility of Cl⁻, $c_{KCl}$ is the KCl concentration in mol L⁻¹, $e$ is the electronic charge, $D$ is the tube diameter, and $L$ is the tube length. If $\mu_K$, $\mu_{Cl}$, and $c_{KCl}$ remain constant during the measurement, we could obtain $G \propto D^2 / L$ that means the variations of $D$ have more prominent effects on the values of SWCNT conductance. Therefore, the narrow conductance range implies a narrow SWCNT diameter distribution after the DGU separation. We followed a method reported in literature to determine the SWCNT diameter by threading polyethylene glycol (PEG) of different molecular weight through the nanopore[22] (Supplementary Fig. 7). We focused on the SWCNTs with conductance in the range of 0.8–1.0 nS, which were used in the subsequent studies. The results in Fig. 1b, c and Supplementary Fig. 7 demonstrate that the diameters of these SWCNTs range from 1.1 to 1.5 nm and center around 1.2 nm.

Next, we measured current-voltage (I-V) curves of the selected range of SWCNT nanopores. It was found that all the I-V curves exhibited linear characteristics when both chambers were filled with the same concentrations of electrolyte (Supplementary Fig. 8). For any of the SWCNT nanopores, when the solution pH was adjusted from 8.0 to 3.0, the conductance gradually decreased presumably due to the protonation of the end carboxylic groups. Then, we further explored the ion selectivity of the tubes with reversal potential measurements in which the two sides of the SWCNT nanopore are exposed to different concentrations of electrolyte (cis 0.1 M KCl/trans 1.0 M KCl). For one example shown in Fig. 1d, the selectivity ratio (SR) of K⁺/Cl⁻ is 2.14 at pH 8.0, which is about the same magnitude as those of biological

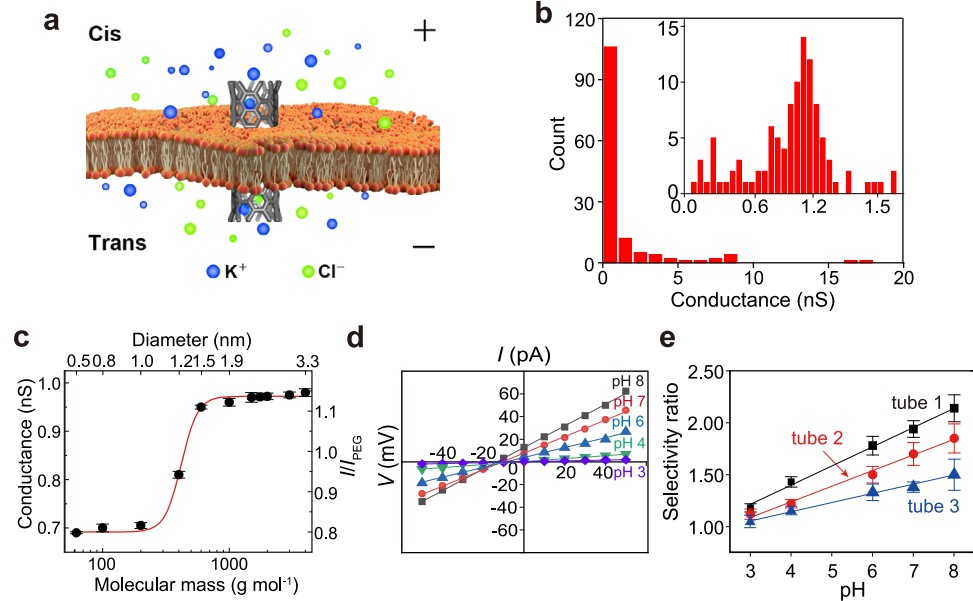

**Fig. 1 | Illustration of the setup and characterization of SWCNT nanopores.**
**a** Schematic of the setup of SWCNT nanopores. **b** Conductance distribution of SWCNT nanopores between 0-20 nS. The inset shows conductance distribution between 0-2 nS. Data were acquired in the buffer of 1.0 M KCl and 10 mM Tris, pH 8.0. **c** SWCNT nanopore blockade as a function of the hydrodynamic diameter of PEG (from 62 Da to 4000 Da). The significant conductance change between PEG 400 and PEG 600 indicates that the diameter of the SWCNT nanopore is between 1.2 nm and 1.5 nm. Data were in the buffer of 1.0 M KCl, 10 mM tris, pH 8.0 under the transmembrane potential of +40 mV. Data are presented as mean values ± SD.

Number of individual experiments $n$ = 3. **d** I-V curves of a SWCNT nanopore under different pH conditions. HCl was added into both sides of the SWCNT nanopore to lower the pH from 8.0 to 3.0, and the conductance dropped from 0.95 nS to 0.04 nS. **e** Ion selectivity ratio (K⁺/Cl⁻) as a function of pH values for three SWCNT nanopores with different conductance (at pH 8.0—tube 1: 0.95 nS; tube 2: 2.12 nS; tube 3: 4.53 nS). Number of individual experiments $n$ = 3. The experiments of (**d**) and (**e**) were conducted in the buffer of trans 1.0 M KCl / cis 0.1 M KCl and 10 mM Tris.

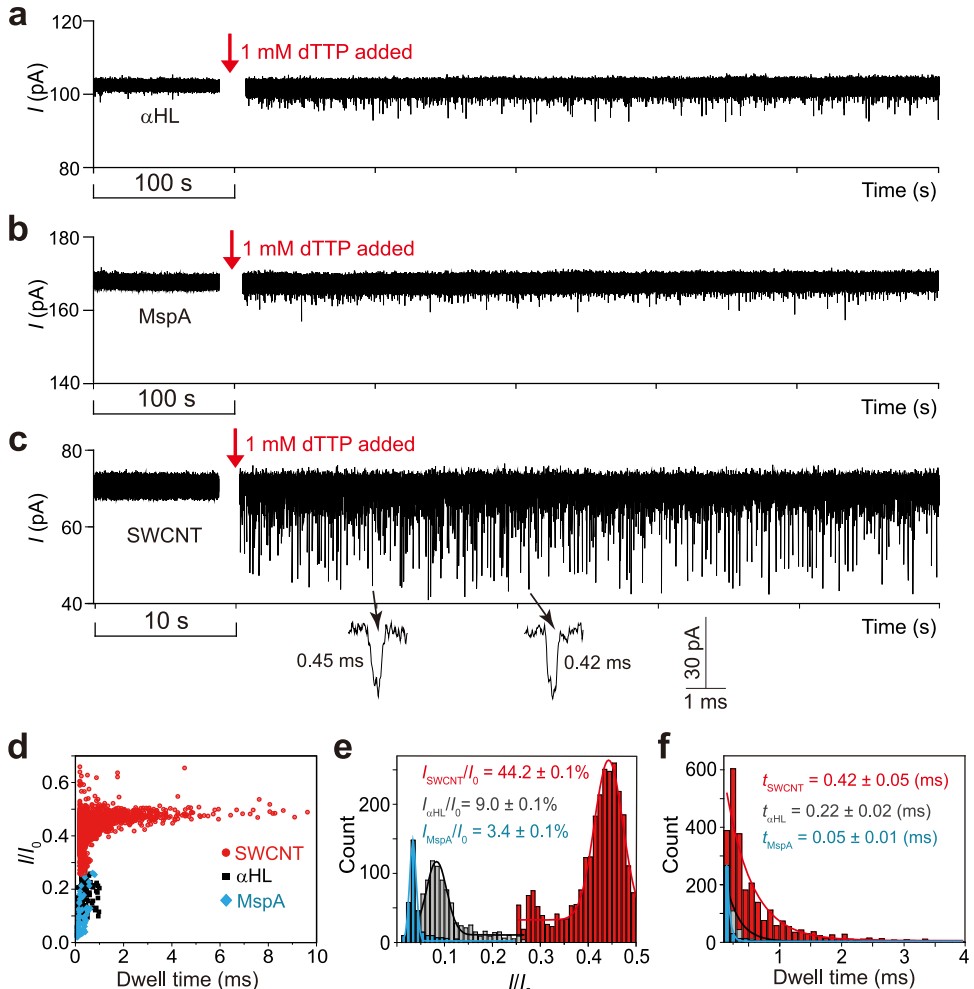

**Fig. 2 | Translocation of dTTP through αHL, MspA and SWCNT nanopore.**
Current traces of the translocation of dTTP through (**a**) αHL, potential held at +100 mV and current 105 pA; (**b**) MspA, potential held at +100 mV and current 166 pA; (**c**) SWCNT nanopore, potential held at +80 mV and current 70 pA (conductance $G$ = 0.83 nS). **d** Scatter plots of the current blockade $I/I_0$ versus event durations in **a**–**c** (SWCNT, circle; αHL, square; MspA, diamond). **e** Histograms of the current blockades of dTTP translocation in αHL, MspA and SWCNT nanopore. (**f**) Dwell time histograms of dTTP translocation in αHL, MspA and SWCNT nanopore. Translocation experiments were conducted in the buffer of 1.0 M KCl, 10 mM Tris, pH 8.0. The sampling frequency is 100 kHz and the filtering rate is 5 kHz.

α-hemolysin (αHL) nanopore and its mutants[23], but much less prominent than that of 0.8-nm-diameter SWCNTs (SR of K+/Cl- ≈ 40) reported by Noy and coworkers[24]. When the solution pH was gradually tuned in situ to 3.0 with hydrochloric acid, we observed the decrease of both the conductance and SR of K+/Cl- (Fig. 1d and Supplementary Table 1). Apparently, the end carboxylic acid groups are the origins of the cation selectivity of SWCNT nanopores. We also tested the SWCNTs of lower fractions and found that the larger conductance tubes have smaller SR of K+/Cl-, which is well in line with previous reports[24]. Similarly, when pH is lowered from 8.0 to 3.0, the SR of K+/Cl- also decreased considerably (Fig. 1e).

## Translocation of dTTP through SWCNT nanopore

To test the sensing ability of these ≈1.2-nm-diameter SWNCT nanopores, we used 2′-deoxythymidine triphosphate (dTTP) as the model target. For comparison, we conducted the dTTP translocation experiments through mycobacterium smegmatis porin A (MspA, see Methods), αHL and SWCNT nanopore, respectively, under the identical conditions. It was found that translocation of dTTP through MspA or αHL only produces a few current events per minute while the capture rate of dTTP in SWCNT nanopore reaches ≈150 events min-1 (Fig. 2, a-c and Supplementary Figs. 9 and 10). The results of voltage-dependent interactions between dTTP and SWCNT nanopore confirmed that dTTP is translocated through the SWCNT (Supplementary Fig. 11). Other than that, the current blockages caused by the presence of dTTP in the three types of pores are significantly different (3.4% vs 9.0% vs 44.2%, Fig. 2d). Considering the size of MspA constriction (1.2 nm) and αHL constriction (1.4 nm) are quite close to the diameter of the SWCNT nanopore (≈1.2 nm), and their length are similar too, the higher capture rate in SWCNT nanopores may result from the distinctive interactions between the π-structure of SWCNT and the aromatic group of dTTP molecule. As for the drastic differences in the current blockages by dTTP, it points to a presumption that ions transport faster in SWCNTs than in MspA and αHL, where ion mobilities are shown to be the same as bulk mobility[25,26]. According to a previously established model[6,27], the conductance changes due to the dTTP translocation through a nanopore, $\Delta G_{pore}$, can be expressed as:

$$\Delta G_{pore} = 6.02 \times 10^{26} \times \frac{1}{L_{pore}} \left( -\frac{\pi}{4} d_{dTTP}^2 (\mu_K + \mu_{Cl}) c_{KCl} e + \mu_K^* q_{dTTP}^* \right) \quad (2)$$

where $L_{pore}$ represents the length of MspA, αHL or SWCNT, $d_{dTTP}$ is the diameter of dTTP, $\mu_K$ and $\mu_{Cl}$ are the electrophoretic mobilities of potassium and chloride ions respectively, $\mu_K^*$ is the effective

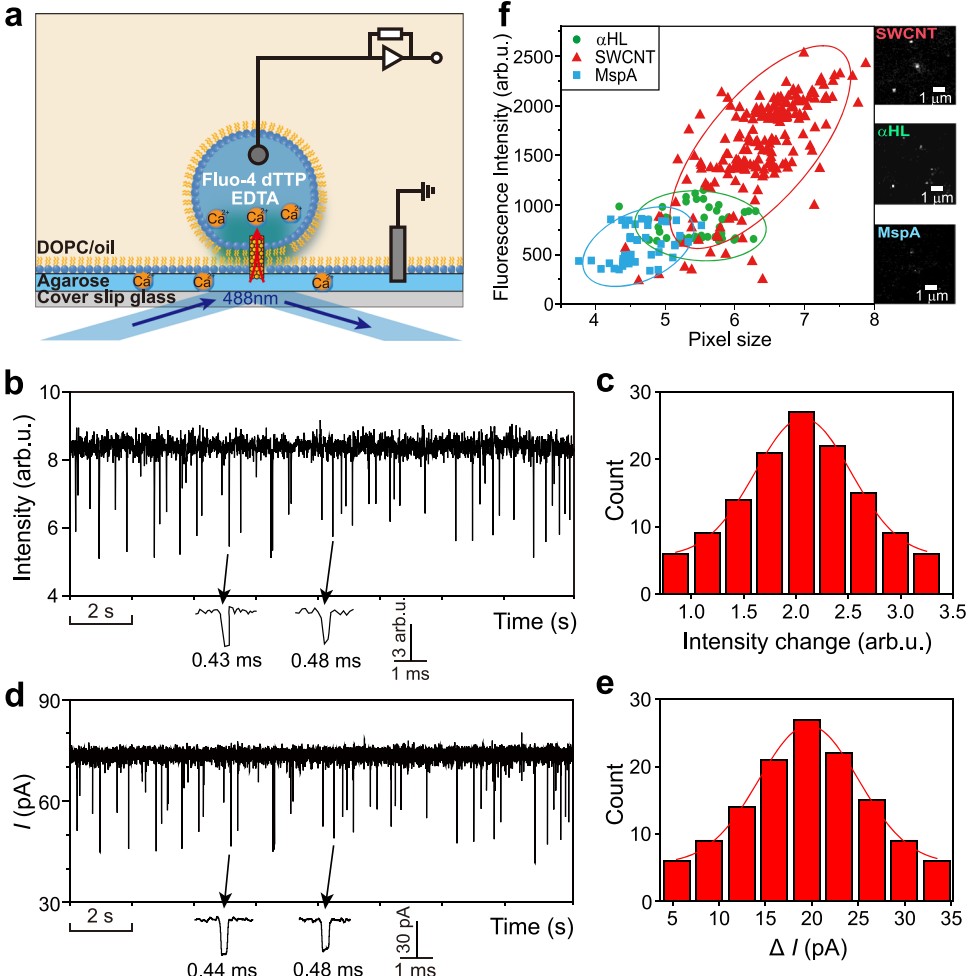

**Fig. 3 | Optical imaging of Ca²⁺ flux through αHL, MspA and SWCNT nanopore in a DHB device. a** Schematic of a DHB device made up with an aqueous droplet and an agarose substrate spun onto a coverslip (grey), both bathed in the DOPC-in-oil solution. In this DHB device, the droplet (trans side) contains the buffer of 1.5 M KCl, 50 μM EDTA, 10 mM Tris, 1.0 mM dTTP, 25 μM Fluo-4 at pH 8.0, and the agarose side (cis side) contains 1.5% (w/v) low melting agarose and 0.75 M CaCl₂, 10 mM Tris at pH 8.0. **b**–**e** Simultaneous recordings of fluorescence intensity changes (**b**, **c**) and ionic current changes (**d**, **e**) caused by the translocation of dTTP through a SWCNT nanopore (conductance $G$ = 0.93 nS). Data were acquired in the buffer of cis side (hydrogel) 0.75 M CaCl₂, 10 mM Tris, at pH 8.0 and trans side (droplet) 1.5 M KCl, 50 μM EDTA, 10 mM Tris, 1.0 mM dTTP, 25 μM Fluo-4 at pH 8.0 with the trans-membrane potential held at +80 mV for both optical imaging and electrical recordings. **f** Scatter plots of the fluorescence intensity and pixel size of the fluorescent dots caused by the Ca²⁺ flux through SWCNT nanopore (triangle), αHL (circle) and MspA (square). Each group of data is circled by an oval with 95% confidence level. The insets show the fluorescent images of the Ca²⁺ flux through the three types of nanopores in 20 ms frames (7.1 × 7.1 μm²) from the image stack. Scale bar, 1.0 μm.

electrophoretic mobility of potassium ions moving along dTTP, and $q_{dTTP}^{*}$ is the effective charge on dTTP per unit length. Although dTTP is a small molecule, its diameter $d_{dTTP}$ is close to the constriction size of MspA and αHL or the diameter of CNT $d_{SWCNT}$. Therefore, the conductance change $\Delta G_{pore}$ caused by dTTP translocation could be estimated mainly by the first term in Eq. 2. Therefore, we can obtain $\Delta G_{pore} \propto (\mu_K + \mu_{Cl})/L_{pore}$, that is $(\mu_K + \mu_{Cl}) \propto \Delta G_{pore}L_{pore}$. If we take the approximation, $\mu_K \approx \mu_{Cl}$, then $\mu_K \propto \Delta G_{pore}L_{pore}$. For MspA, $L_{MspA} = 10.0$ nm, $\Delta G_{MspA} = 0.056$ nS; for αHL, $L\alpha_{HL} = 10.0$ nm, $\Delta G\alpha_{HL} = 0.095$ nS; whereas for SWCNT nanopore, $L_{SWCNT} \approx 5.0$-$15.0$ nm, $\Delta G_{SWCNT} = 0.385$ nS (Fig. 2d and Supplementary Fig. 5a). By comparison of these data, we could conclude that the ion mobilities $\mu_K$ or $\mu_{Cl}$ in SWCNT nanopores are 3-5 times higher than that in biological nanopores where ion mobilities are shown to be the same as bulk mobility[25,26]. This agrees well with our previous study of DNA translocation in larger-diameter SWCNT nanopores[6] and a study of ion mobilities in SWCNT membranes[28]. This conclusion is also in qualitative agreement with a molecular dynamic simulation study by Peter and Hummer[26], which showed an ≈50% increase in ion mobilities inside short SWCNTs.

## Optical imaging of Ca²⁺ flux through SWCNT nanopore

To further corroborate the finding of faster ion transport in SWCNT nanopores, we performed optical imaging of calcium ion (Ca²⁺) flux through SWCNT nanopore, αHL, and MspA, respectively. Previous measurements of the flow of gases, liquids, and ions through CNTs driven by either pressure or electrical field were mostly based on CNT membranes or CNT-inserted liposomes[2-4,24,28,29]. The results of the flow rates are the average of many heterogeneous CNTs. In this work, our setup allows us to focus on the measurements of cation transport inside individual single SWCNTs. The ion selectivity experiments indicate that the SWCNT nanopores are moderately cation selective, which means there are still considerable amount of chloride ions flowing through the tubes. By using total internal reflection fluorescence (TIRF) microscopy of droplet hydrogel bilayers (DHBs) (see Methods), we are able to image the Ca²⁺ flux through different types of nanopores (Fig. 3a and Supplementary Figs. 12 and 13)[30,31]. When Ca²⁺ in the hydrogel chelates Fluo-4 (Ca²⁺-sensitive dye) in the droplet on the droplet-hydrogel interface through the nanopore, the fluorescence of Fluo-4 would be enhanced by hundreds of times regardless of the presence of Cl⁻. We first tested the imaging system with αHL nanopore

and a molecular adapter γ-cyclodextrin (γCD). The results showed that the fluorescent signals of the reversible binding of γCD inside the lumen of αHL nicely matched the ionic current signals of the same process (Supplementary Fig. 14). Next, we employed the system to monitor the Ca²⁺ flux through SWCNT nanopores. When a SWCNT was inserted in the lipid layer, Ca²⁺ in the hydrogel started to migrate toward the droplet through the SWCNT. After translocation, Ca²⁺ met Fluo-4 at the cis mouth of the SWCNT and then the enhanced fluorescence of Fluo-4 could be visualized and recorded. In this case, we put dTTP in the droplet together with Fluo-4. As a result, we observed blinking of the fluorescent signals that was due to the Ca²⁺ ions passage blocked by dTTP and further confirmed the insertion of SWCNTs (Fig. 3b, Supplementary Fig. 15, and Supplementary Movie 1). These fluorescence fluctuations correspond well to the ionic current recordings (Fig. 3c). Finally, we substituted the SWCNT nanopore with MspA and αHL in the imaging system, and recorded optical signals of the Ca²⁺ flux through these two biological pores (Supplementary Movies 2 and 3). The statistical results of Ca²⁺-enhanced fluorescence of Fluo-4 through SWCNT, MspA, and αHL are shown in Fig. 3d. It is interesting that although the variations in fluorescence data are slightly larger than electrical current recordings, the distribution of maximum fluorescence intensity of Ca²⁺-Fluo-4 in the three types of nanopores remarkably resembles the conductance changes in these pores in Fig. 2e. Because the diameters of SWCNT, MspA, and αHL are very close (1.2–1.4 nm), and the intensity of Ca²⁺-enhanced fluorescence is proportional to the local concentration of Ca²⁺, the exceptionally high fluorescence intensity caused by SWCNT-transported Ca²⁺ could only be accounted for by the higher electrophoretic mobilities of Ca²⁺ inside SWCNTs.

### Discrimination of amino acids in SWCNT nanopores

Higher ion mobilities inside SWCNT can lead to more prominent current changes during stochastic sensing. This distinctive feature prompted us to develop high-resolution nanopore sensors based on SWCNTs. Protein sequencing by nanopores is a burgeoning area, but there is no consensus on how this could be eventually achieved. Very recently, three groups independently demonstrated that a short peptide conjugated to a DNA strand could be read by a nanopore at the single amino acid resolution with the aid of DNA helicase/polymerase motors[31–33]. However, each recorded current level contains the contributions of an "8-mer" of amino acids, and currently it is not possible to resolve the astronomical number of combinations of amino acids[34,35]. Therefore, de novo sequencing with this approach is still beyond reach. An alternative way of enzymatic cleavage followed with discrimination of amino acids holds some promise of success. Apparently, either strategy would be very challenging and disruptive, but we will only focus on the discrimination of proteinogenic amino acids with SWCNT nanopores in this work. There are twenty proteinogenic amino acids and their sidechains vary in size, charge and hydrophobicity, etc. Among them, the pairs most difficult to differentiate are homologue amino acids and chain isomers. To tackle this challenge, we chose three pairs of amino acids and conducted their discrimination with SWCNT nanopores. The first pair is aspartic acid (Asp) and glutamic acid (Glu), among which there is only a methylene difference. We first placed Asp in the cis chamber and measured its translocation events through a SWCNT nanopore ($G$ = 0.83 nS). We observed short and transient spike events. Due to the uncontrollable heterogeneity of SWCNTs, we used the same tube for the sensing of Glu but placed it in the trans chamber instead under negative transmembrane potentials. After the inversion of the signals, we found that the events produced by Glu exhibit slightly longer duration and much deeper current blockage. From the statistical data shown in Fig. 4a, it is clear that Asp and Glu could be completely differentiated by the SWCNT nanopore. Similarly, phenylalanine (Phe) and tyrosine (Tyr) that differ only by one hydroxyl group could also be discriminated in a

SWCNT nanopore (Fig. 4b). The most remarkable example is leucine (Leu) and isoleucine (Ile), the pair of chain isomers that have never been reportedly discriminated by any nanopore sensors. Although they have the same molecular formula and differ only in skeletal structures, their translocation profiles through SWCNT nanopore, e.g., current blockage and capture rate, are markedly different (Fig. 4c). The drastic differences in current blockage caused by the pair of isomers could readily tell them apart. To eliminate the possibility of discrimination arising from the SWCNT entrance differences, we repeated all the above experiments and finally placed the pair of amino acids in the same chamber. The results showed that discrimination of the mixture is readily acquired and there are no entrance differences (representative examples in Supplementary Figs. 16–18). It should be noted that currently, we cannot run all the amino acid discrimination in one single SWCNT nanopore. The heterogeneity of SWCNTs is considered as a disadvantage in the sensing applications, but it also provides opportunities for building a sensing array in parallel based on different SWCNT nanopores to capture and distinguish all the twenty amino acids, which could serve as a critical component of a peptide sequencing system.

## Discussion

In summary, we have developed a modified approach to separate acid/sonication-cut SWCNTs with density gradient ultracentrifugation and employed the ultrashort SWCNTs (≈5–10 nm) to construct nanopore sensors inserted in lipid bilayers. These SWCNT nanopores exhibit conductance values in a relatively narrow range and have high sensitivity in sensing small molecules such as dTTP. The conductance changes during dTTP translocation through SWCNT nanopores are 3–5 times higher than that in biological nanopores such as MspA and αHL. We attribute this unusual phenomenon to faster ion transport in SWCNT nanopores than in bulk solutions, which is corroborated by the optical Ca²⁺-Fluo-4 imaging experiments on DHBs. Furthermore, we apply SWCNT nanopores to the discrimination of proteinogenic amino acids. The results show that for the most challenging three pairs of amino acids-Asp and Glu, Phe and Tyr, Leu, and Ile, sufficiently high resolution is achieved to differentiate each pair.

To develop a complete peptide sequencing system based on enzymatic cleavage strategy, there are three major hurdles that must be overcome. The first is to engineer an enzyme that can continuously digest the peptide one amino acid a time, ideally at a constant rate. The second is to build a SWCNT nanopore array that can unequivocally discriminate all the twenty proteinogenic amino acids. The third is to construct the enzyme-SWCNT hybrid that ensures catching and reading of each cleaved amino acid. Our work in this report is a small but significant step toward a long-awaited, advanced protein sequencing technology.

## Methods

### Materials and characterization

1,2-Diphytanyol-sn-glycero-3-phosphocholine (DPhPc) and 1,2-dioleoyl-sn-glycero-3-phosphocholine (DOPC) were purchased from Avanti Polar Lipids (Alabaster, AL). The long SWCNTs (300 nm–4.0 μm) solution was purchased from NanoIntegris Technologies Inc. (USA). The dTTP sample was purified by HPLC and obtained from Sangon Biotechnology (99.9%, Shanghai). The 3 kDa centrifugal ultrafiltration tube was purchased from Millipore Corp. (Billerica, MA, USA). Chemical reagents were purchased from Sigma-Aldrich (Merck, USA). Fluo-4 penta-potassium salt for the fluorescence intensity experiments was purchased from Beyotime (99.9%, Shanghai). Density gradient ultracentrifugation was conducted on XPN-100 ultracentrifuge (Beckman coulter, USA). TIRF measurements were performed on a microscope (Nikon Eclipse Ti2, Japan) equipped with an oil immersion objective (Nikon 100x Plan Apo TIRF, Japan). Fluorescence was excited by a 488 nm Argon ion laser (4mW OBIS, Coherent,

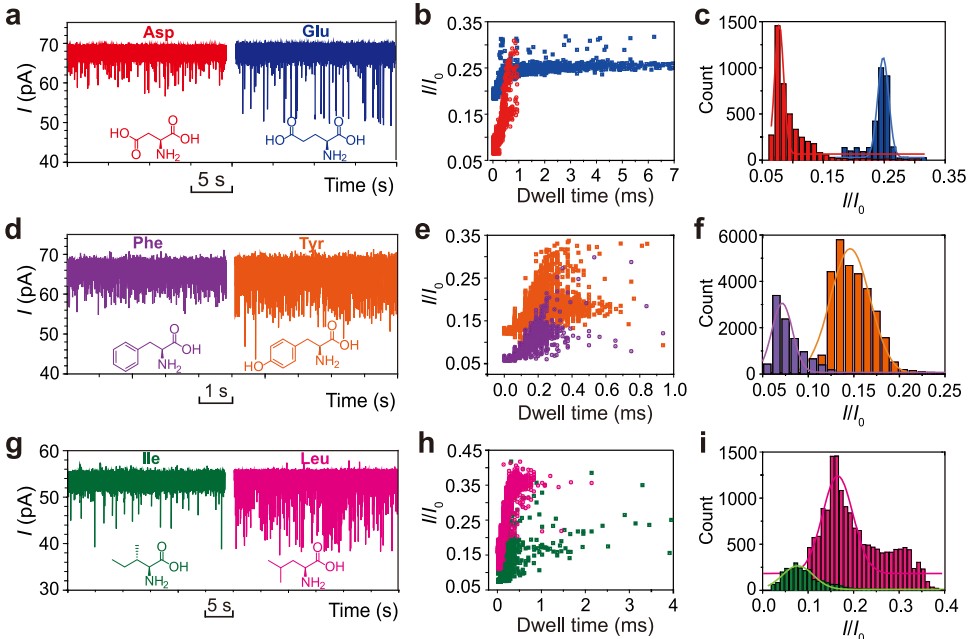

**Fig. 4 | Discrimination of three pairs of amino acids using SWCNT nanopores.**
**a–c** Typical current traces, scatter plots of the events and histogram of normalized current blockage $I/I_0$ of the translocation of Asp (red) and Glu (blue) through a SWCNT nanopore ($G = 0.83$ nS). The final concentration of Asp and Glu is 500 μM. The transmembrane potential was held at 80 mV and the current was 68 pA.
**d–f** Typical current traces, scatter plots of the events and histogram of normalized current blockage $I/I_0$ of the translocation of Phe (purple) and Tyr (orange) through a SWCNT nanopore ($G = 0.94$ nS). The final concentration of Phe and Tyr is 200 μM.

The transmembrane potential was held at 70 mV and the current was 69 pA.
**g–i** Typical current traces, scatter plots of the events and histogram of normalized current blockage $I/I_0$ of the translocation of Ile (green) and Leu (magenta) through a SWCNT nanopore ($G = 0.87$ nS). The final concentration of Ile and Leu is 200 μM. The transmembrane potential was held at 60 mV and the current was 54 pA. Data were acquired in the buffer of 1.0 M KCl, 10 mM Tris and pH 8.0. All the experiments were successfully repeated more than 10 times.

USA) and imaged with an electron-multiplying CCD camera (Ixon L897, Andor, UK). The transmission electron microscope (TEM) imaging was carried out on a HRTEM Tecnai F20. The atomic force microscope (AFM, Bruker Dimension icon) was used to perform length measurements of the ultrashort SWCNTs. The ultrashort SWCNTs were measured in tapping mode using super-sharp AFM tips (SuperSharpSilicon™ tips from Nano Sensors; tip radius 2.0 nm).

### Protein preparation
Wild-type αHL was produced by expression in BL21 (DE3) pLysS *Escherichia coli* cells. The monomers were then assembled into homoheptamers on rabbit red blood cell membranes followed by purification with 8% SDS-PAGE. The purified heptamer protein was conserved in buffer (10 mM Tris-HCl, pH 7.9, 50 mM NaCl) and stored at -70 °C[36].

The gene coding for the monomeric M2-NNN MspA mutant (D93N/D91N/D90N/D118R/D134R/E139K, designated as MspA in this work) was synthesized and inserted in a pet-30a(+) vector. The M2 MspA was expressed with *Escherichia coli* BL21 (DE3) and purified using nickel affinity chromatography (GE Akta Pure, GE Healthcare). The purified M2 MspA spontaneously oligomerizes into an axis symmetric, octameric form, ready for single-channel recording experiments[37].

### Preparation of short SWCNTs
To cut the long SWCNTs (300 nm–4.0 μm) into ultrashort SWCNTs (5–10 nm), 20 mg long SWCNTs was dispersed into the mixture of 24 ml $H_2SO_4$/$HNO_3$ (3:1), and then sonicated in a water bath (800 W) for 50 h at 35–40 °C. The mixture was then diluted 10 times and neutralized with 6 M NaOH solution. The longer portion of SWCNTs were filtered out by a PTFE filter (0.22 μm; Millipore) and recycled for sonication. The shortened SWCNTs solution was then centrifuged with 3 kDa centrifugal ultrafiltration tube, and washed with water for 3 times.

### Separation of short SWCNTs with density gradient ultracentrifugation
The process of density gradient ultracentrifugation to separate different length of SWCNTs was modified based on the procedure published previously[21,38]. In brief, the OptiPrep solution (Iodixanol, 60% (w/v), Sigma-Aldrich Inc.) was first diluted into 5%, 7.5%, 10% concentration with 2% (w/v) sodium dodecyl benzene sulfonate (SDBS, Aladdin, Shanghai) and 1.5% (w/v) deoxycholic acid sodium salt (DOC, Aladdin, Shanghai). The ultracentrifuge tube was filled with different concentrations of OptiPrep (iodixanol concentration from 5% to 10%) and to the top of that was loaded the SWCNT solution. Then the mixture was ultracentrifuged for 8 h under 4 °C and 288,000 × *g* conditions with a SW 41 Ti swing bucket motor. Finally, the SWCNT solution was separated into Eppendorf tubes (200-400 μL) layer by layer.

### Raman spectra
The SWCNTs samples were prepared by dropping the solution of different fractions onto a glass slide which was then dried by an electrical heater. The Raman measurements were performed at 1.96 eV (632.8 nm) excitation at ×20 magnification and 5 μm spot size (Renishaw, RM2000). Laser power of 4.7 mW was used to prevent destruction of the samples during measurements. Raman spectra showed that the shortened SWCNTs had very few or no radial breathing mode (RBM) band and after the DGU process the RBM band reappeared, which indicated that the DGU process indeed purified the SWCNTs by density.

### Calculation of ion selectivity of SWCNT nanopores
To reduce the effect caused by liquid junction potential, the salt bridge (1.0 M KCl and 5% agarose) was used. In these cases, both electrodes were dipped in a solution of 1.0 M KCl, 10 mM Tris, at pH 8.0. Then, the reversal potential of SWCNT nanopores was measured. Theoretical

redox potential values were calculated using the Nernst equation:

$$\triangle E = \frac{RT}{F} \cdot \ln \frac{a_{cis}}{a_{trans}} \qquad (3)$$

where $\triangle E$ is the theoretical potential offset observed at the electrodes, R is the gas constant, $T$ is the temperature of the solution, F is the Faraday constant, and $a$ is the activity of the ion in solution. The reversal potential was then used with the Goldman-Hodgkins-Katz equation (Eq. 4) to calculate the transference numbers for the membrane[24,39].

$$V_R = (2T_{K^+} - 1)\frac{RT}{F} \cdot \ln \frac{a_{cis}}{a_{trans}} \qquad (4)$$

where $V_R$ is the measured reversal potential, and $T_{K^+}$ is the effective transference number of potassium ion through the SWCNT nanopore within the membrane. With knowledge of the effective cation transference number for the membrane, the ionic selectivity ratio, SR (cation/anion), was calculated using Eq. 5.

$$SR = \frac{T_{K^+}}{1 - T_{K^+}} \qquad (5)$$

### DHB device

A DHB device was built according to the previous literature reports by integrating an agarose-coated glass coverslip into a polymethylmethacrylate (PMMA) device (created by computerized numerical control machine, Supplementary Fig. 12) suitable for TIRF illumination[30,31]. First, a molten layer of agarose (1.5% w/v in water) was sprayed onto the coverslip glass and then covered by DOPC mixture (5 mg mL$^{-1}$ DOPC in hexadecane oil). Second, a PMMA device was dried by the nitrogen gas flash and filled with molten agarose (3.5% w/v, experimental buffer solution) in the channels. Third, the coverslip glass was sealed to the PMMA device and DOPC mixture was filled into the 1.5 mm holes of the PMMA device. Finally, the droplet (contained experimental buffer solution bathed in the DOPC mixture) was settled into the 1.5 mm holes of the PMMA device which could be controlled by a micromanipulator.

### Fluorescent imaging of Ca$^{2+}$ flux through MspA/αHL/SWCNT

DHBs were imaged with a 100× objective (Plan Apo TIRF; Nikon Instruments) excited with a 488 nm Argon ion laser (Coherent OBIS, 4mW). Fluorescence was detected by an electron-multiplying CCD camera (Ixon L897). The images and videos were analyzed with both NIS Elements D Analysis and ImageJ software[40], and the particle tracker plugin developed by the Computational Biophysics Lab at ETH Zurich[41]. The presence of target nanopores (red circles in Supplementary Fig. 15) were confirmed by fluorescence blinking experiments and the fluorescence intensity of the brightest frame were measured for each nanopore. For the three types of nanopores, 95% confidence ovals were produced for the relationship between fluorescence intensity and pixel size. Minimum time scale is 20 ms for CCD camera because the maximum frame number of the CCD camera is 50 in ROI (region of interest, 7.1 × 7.1 μm$^2$).

### Single-channel current recording

DPhPC was used to form a stable lipid bilayer across an aperture 100 μm in diameter in a 25-μm-thick Teflon film (Goodfellow, Malvern, PA) that divided a planar bilayer chamber into two compartments, cis and trans. Both compartments contained 1 mL of KCl/CaCl$_2$ buffer solution. Samples were added in the cis compartment, which was connected to the ground. The trans side was connected to the headstage of the amplifier. Ionic currents were measured by using Ag/AgCl electrodes with a patch-clamp amplifier (Axopatch 200B; Axon

instruments, Foster City, CA), filtered by a low-pass Bessel filter with a corner frequency of 5 kHz and then digitized with a Digidata 1440A A/D converter (Axon Instruments) at a sampling frequency of 100 kHz. The transmembrane potentials were mentioned in the article.

### Data analysis

Current traces and signal events were analyzed with Clampfit 10.2 software (Axon Instruments). Events were analyzed by using single channel search, and used to build current change and dwell time scatter diagram. ImageJ software was used to track the nanopore position and fluorescence intensity. Origin software (Microcal, Northampton, MA) was used to analyze the fluorescence intensity and ionic current data for histogram construction, curve fitting and graph presentation purposes.

### Reporting summary

Further information on research design is available in the Nature Portfolio Reporting Summary linked to this article.

## Data availability

The data that support the findings of this study are available from the corresponding author upon request. Source data have been deposited in the Zenodo database under accession code: https://doi.org/10.5281/zenodo.7874404[42].

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

## Acknowledgements

This project was funded by the National Natural Science Foundation of China (no. 21874136 to L.L. and nos. 22025407, 21974144 to H.-C.W.), and Institute of Chemistry, Chinese Academy of Sciences.

## Author contributions

W.P. performed SWCNT preparation and characterization experiments; W.P. and S.Y. performed single-channel recording experiments; K.Z. provided protein nanopores; W.P., H.-C.W. and L.L. performed data analysis; H.-C.W., L.L. and Y.Z. conceived the project, designed the experiments; W.P, H.-C.W. and L.L. wrote the paper.

## Competing interests

The authors declare no competing interest.
