## [Peer Review File · Nature Communications]

High-resolution discrimination of homologous and isomeric proteinogenic amino acids in nanopore sensors with ultrashort single-walled carbon nanotubesReviewers' Comments:

Reviewer #1:

Remarks to the Author:

The report by Peng et al is the best quality CNT-based translocation sensing data out there and is based on adding an Density Gradient Ultra-centrifugation(DGU) purification step that has worked well for CNT gradation by size/chirality (Hersam group Northwestern Univ. ref 34). However, I hesitate to recommend it for publication in a top tier journal. The major detractors of the work are:

The main advance was to add a purification step to get the most uniform CNTs possible for the translocation experiments. However there was very little characterization of the CNTs outside of Raman, size exclusion series, and the conduction experiments themselves (inference only of CNT L&D). TEM characterization is desperately needed. The presence of RBM is encouraging, but Fig S3 c is significantly higher signal than b, and RBM is higher than G. So a few inconsistencies. Most of the data shown appears to come from one or a few different samples. If the primary advance of this work is a more uniform pore made possible by DGU, then a statistically valid number of different samples are needed.

They quote slightly enhanced mobilities (2-3), but conductance has a D^2/L dependence. These values have a high degree of uncertainty, thus it is not appropriate to report them without uncertainty calculated. I suspect the uncertainty will be on the order of the calculated value, not really appropriate for a top journal.

Fig S4 has quoted diameters of 1.7nm and 1.9nm for two different samples. Manuscript shows 1.2nm. The primary hypothesis of uniformity is not supported.

The title has 'Atomic resolution sensing..' however this is not at all proven. The diameter control is not demonstrated to the angstrom scale precision. The molecules sensed (amino acids) differ by more than a few atoms. Asp vs Glu has an extra CH₂ link, giving more conformational flexibility. 'Molecular sensing...' would be more appropriate.

Fig 4 is the most compelling one, showing a distinction between amino acid monomers. It is not clear if the same trend would be repeated with different CNT diameter (i.e. different samples). Short peptide chains (~7-10 units) of programmed sequence can be easily purchased. This is the more appropriate target for sensing.

Minor points: The abstracts first few sentences are almost contradictory. Are CNTs near perfect or highly inhomogeneous? Make more clear what the defects are. The introduction reads as a competition between groups, best to leave group names out and focus on the advancement of the technology in the field. Ref. 20 should be explicitly cited as the first group to purify SWCNTs via DGU. Presently it is buried referring to supplemental figures.

Reviewer #2:

Remarks to the Author:

The submission by Peng et al describes the preparation of single-walled carbon nanotubes (SCWNTs), their in-depth characterization in terms of unique conductance properties, and applications for single-amino acid discrimination which is of relevance for label-free sensing applications.

The preparation involved density gradient ultracentrifugation to obtain up to several nm-short tubes of narrow width distribution in the range of 1 nm. This has so far not been used and obtained.

The chemical nature of the pores was analyzed with Raman spectroscopy, their width with probe molecules of various sizes in electrical recordings, and their ion selectivity and pH-dependent ion

conductivity to confirm the influence of diameter and terminal COOH groups. Using electrical recordings and optical imaging of transport, the pores were found to show unusually high conductance which was attributed to the high mobility of electrolyte ions in the SCWNTs.

The high conductance of the tubes was applied to distinguish amino acid pairs. The clear distinction between structurally similar leucine vs isoleucine residues is striking.

The manuscript could be of interest for Nature Communications given the newly applied approach to obtain defined SCWNTs, the thorough characterization of the tubes (exception see below), and the exploitation of the unique conductance properties for a previously very difficult to achieved discrimination of analytes via nanopore sensing. The manuscript is clearly structured and generally well written.

Before the manuscript can be considered for publication, the following points should be addressed.

Major:

Length characterization: The authors claim a length of 5-10 nm for the carbon tubes. But the length of the tubes was not experimentally determined. This should be done such as via AFM.

The experimentally determined higher conductance of carbon nanotubes should be quantitatively compared with existing data on other, longer carbon nanotube conductance. The higher conductance should be explained by referring and updating biophysical models or potentially simulations which have been developed previously.

The units for equation 1 are strange

G should be S or $\text{kg}^{-1} \text{m}^{-2} \text{s}^3 \text{A}^2$

$D^2(uK)c(KCl) e$ is $\text{m}^2 \text{V}^{-1} \text{s}^{-1} \text{mol L}^{-1} \text{m}^2 \text{m}^{-1}$ or $\text{m}^3 \text{kg}^{-1} \text{m}^{-2} \text{s}^3 \text{A mol L}^{-1}$

Giving

$\text{kg}^{-1} \text{m}^{-2} \text{s}^3 \text{A}^2 = \text{m}^3 \text{kg}^{-1} \text{m}^{-2} \text{s}^3 \text{A mol L}^{-1}$

Giving a proportionality factor with units of

$\text{m}^{-3} \text{mol}^{-1} \text{L A}$

Minor points:

Page 3: The historic account is one way of introducing the topic but scientifically likely not the most exciting. It could be better to restructure the first introduction paragraph to follow scientific concepts and ideas.

Page 6, line 114: Should the pH-dependence not follow a sigmoidal dependence similar to typical titration experiments?

Page 7, line 132: Define the distinct sensing mechanism.

Page 9: Why was the blinking observed? Likely as dTTP blocked passage of the Ca^{2+} ion. This should be explained.

Page 10, line 193: The authors ignore the recent break-through work by Dekker et al in Science on nanopore-mediated peptide sequencing. This paper should be cited. The authors' motivation into the amino acid sensing via their carbon nanotube has to be edited accordingly.

Page 11, line 225: Distinguishing amino acids via multiple carbon nanotubes of different selectivities can in my view only be applied for peptide sequencing when the nanopores are in a serial array. But this has so far not been achieved. The authors likely refer to a parallel array of different nanopores

and amino acid selectivities which is very difficult to perceive to achieve sequencing. The paragraph should be edited to spell out more clearly what and what cannot be achieved with their pores in terms of peptide sequencing.

The conclusions chapter is weaker than the results section and should be brought to a higher level. It should feature a more expanded comparison to state of the art, a comparison of the pros and cons of carbon nanotubes for sensing, and more detailed plans on how to develop the technology further.

The formatting of the references is inconsistent.

Reviewer #3:

Remarks to the Author:

Weichao Peng and coworkers isolated ~5-10 nm SWCNT and reconstituted into lipid bilayers. The nanotubes were capable of translocating(?) amino acids, and differences with similar amino acids were identified by current recordings. This is an important work, as the mechanism of molecular recognition in nanopores is unknown, and especially in SWCNT-nanopores. Here, the observation that molecules as small as individual amino acids can be distinguished is intriguing, as the inside of a CNT is chemically uniform. Hence, this work would provide important additional data to understand the recognition mechanism. Furthermore, the ability of distinguishing enantiomeric molecules with nanopores will most likely have important applications.

It will be important, however, to prove whether or not the analytes enter the CNT. The authors should, therefore, provide a voltage dependence of the dwell times for the molecules analyzed. If the dwell time decreases with the bias, then this can be taken as a proof of molecular translocation.

Another point concerns the ability of these CNT to distinguish between similar molecules. As far as I understood, the authors added one analyte on one side of the CNT and another on the other side, and they observe a difference between the two molecules. Why? Could this asymmetry be explained by the fact that the two molecules interact with the entry of the CNT rather than translocating through it. If they have not done so, the authors should add both analytes on the same side and observe the two distributions.

Other points

Line 46. What is a "homemade device"?

Line 51-52. What is the difference between current blockades and spikes?

Line 54. Please indicate what are the difficulties on getting the SWCNT.

Line 80. The connection with protein sequencing is not obvious.

Line 82. Please define what is DGU separation and what is the advance compared to previous protocols used.

The authors should discuss why they have such a large variability in the range 0 - 2 nm. It should be noted that, compared to biological nanopores, this distribution is actually rather wide. Is this due to the different length of the CNT? Or different diameters? Further it appears that there are several distributions (0.2 - 0.4 - 0.7 - 0.9)

Line 89. Where does the equation come from (reference)? The equation should be re-drawn, as it is also not easily understandable.

Line 96. "Literature method" please rephrase

Line 98. The author mention they use the CNT in the range 0.8-1 nS. However, in Fig. 2C and Fig 4 the SWCNT the distribution appears larger.

Line 99. "The results demonstrate...". Which results? Not clear to me what are the authors referring to. Figure 1e. What do Tube 1-2-3 refer to? Three different CNT? If so, why there are error bars?

Figure S5. Why there are no error bars? How many CNT were tested at each pH?

Line 105. Please indicate where to find the IV curves.
Line 112. Please indicate the ion selectivity of the 0.8 nm CNT
Figure 2c. The filtering rate is 5kHz. Why? Can the author use higher filtering? In addition, the authors should add a better representation of individual blockades (more blockades and a better expansion). Why in figure 2c the noise of the experiment increases after adding the sample? Finally, the authors should indicate sampling and filtering in the legend.
Line 122. Is there any reason why the authors used dTTP? In addition, the authors should perform a voltage dependency to check whether the molecules translocate across the nanopore. This is important, because if the molecule do not translocate it will reveal an important information about the mechanism of molecular identification.
Line 127. The authors say: "Other than that, the current blockages caused by the presence of dTTP... " what do they mean? Do they refer to the Ires? How was defined?
Line 131. Is the capture rate higher because the events are longer (missing events)?
Line 135. "... bulk mobility". Please add a reference.
Lines 144-147. Please indicate the ionic strength and applied bias for all the conductance.
Line 147. The range 5-10 nm for the CNT is rather large. Would this affect the ability of the CNT to detect molecules? How can the author distinguish between length and diameter of the CNT if they only select based on the conductance of the CNT?
Line 197. Is sequencing by enzymatic cleavage proposed in this work? If it is, the authors should elaborate more (e.g. how can this be done?). Otherwise, they should add a reference (and perhaps still elaborate). It is not obvious how measuring amino acids can lead to protein sequencing.
Line 209. Please add all the analytes in the same chamber.
Line 2017. What is a 'skeletal' structure?
Line 2018-2019. Please explain the "difference" in the translocation profiles

Reviewer #4:

Remarks to the Author:

Peng, Liu, Wu and co-workers report ion transport and small molecule blockade measurements in short carbon nanotubes of ~1.2 nm diameter inserted into as lipid bilayer matrix. Ion conductance measurements show that ion electrophoretic mobility shows modest enhancement of 2-4x in these pores. The authors also report combined optical and ion transport experiments showing Ca²⁺ ion translocation in CNT pores using a droplet interface bilayer setup. Finally, the authors report that CNT pores can discriminate between several pairs of amino-acids that are often hard to discriminate with remarkable efficiency. Even though the results, and particularly the demonstration of Ca²⁺ ion transport and the amino-acid pair discrimination detection is impressive, this paper raises a number of questions that do not allow me to endorse it, at least in the current form.

Specifically, I find the central result of the unusually strong discrimination of the three pairs of amino-acids by the short carbon nanotube channels interesting (although that part lacks a critical control experiment). I find the mechanistic explanations offered by the authors much less convincing. The other part of the manuscript that reports slightly enhanced electrophoretic mobility of the salt ions in the channel is also not convincing and relies on questionable analysis. The Ca²⁺ ion transport measurements are elegant and fascinating, but they do not seem to connect to the rest of the manuscript narrative and results (which were taken using a different experimental setup). Perhaps refocusing the manuscript on a coherent linear narrative around amino-acid pair discrimination, adding critical controls, and presenting more robust statistics would improve the impact of the work. Some specific comments follow:

1. The connection between "special feature" of the nanotube pores and the enhanced sensing sensitivity discussed on the page 4 is questionable. In fact, this theory of the authors goes more or less against the conventional wisdom of the nanopore field. Interactions of the sensing molecule with the nanopore wall affect the duration of the blockade, but not the blockade amount, which is largely

determined by excluded volume of the molecule. From that point of view a CNT will have a strong disadvantage as a nanopore discriminator over the more conventional pores such as MspA or α -haemolysin (α -HL) that have a defined constriction in the channel that makes for an effectively shorter sensing region. No argument is provided for why authors' conclusion should be correct. In fact it is likely wrong.

2. The same discussion on page 4 also states that faster ion transport in CNTs would lead to higher signal and higher resolution in small molecule translocation studies. It is hard to see how one follows from the other. Even if ions flow faster in the CNT pore, the small molecules would also flow faster, which would only result in shorter duration blockades, unless the authors provide a mechanism for why those small molecules are slowed down.

3. The manuscript claims a narrow conductance range for the short CNTs that they synthesized. Figure 2B shows a rather wide range (and rather sparse statistics with many histogram bars at 1 even count). An average conductance number and an error bar would be helpful in evaluating the quality of the manuscript's statement.

4. What was the initial range of diameters for the long CNT stock? This information may be helpful in evaluating the quality of the purification process.

5. The data on the Fig 2 a-c need to be presented on the same scale to make a fair comparison. The data also show that the absolute level of current blockade by dTTP is roughly similar between the CNT and MspA (about 20 pA), but is much smaller for the case of α -HL (roughly 5 pA). Intuitively, you would expect a similar absolute current blockade value, as it is related to an excluded volume of the molecule being transported. Can the authors explain this discrepancy?

6. The use of equation 1 on page 7 to interpret the experiments on dTTP is incorrect. Even though dTTP is a relatively large molecule, its size is still much smaller than the length of the pore. The model of Smeets and Dekker (Eq.5 in their paper is nearly identical to Eq. 1) was derived for translocating long DNA through the channel where DNA occupies the whole channel length, which is quite different from a case of small molecule translocating through a longer channel. How do the authors justify using the same $1/L$ factor in their equation?

7. The comparison of conductances at the bottom of page 7 borders on disingenuous. Even if we ignore that MspA and α HL pores are not cylindrical and have a pronouncedly conical shape, it looks like the authors are arbitrarily assuming the length of their channels to be the lower boundary of their length estimates (5 nm) to obtain a better agreement with their enhancement factors. What are the errors associated with the measurements and do they even allow for a solid claim of the enhanced mobility in these channels?

8. Figure 3C needs to show a zoomed-in section to show a correlation between the optical signal and ionic current images to make it easier to see the correlation. I had to blow it up and use a ruler.

9. Fig. 4 data: Even though CNT channels are symmetric inside, the two entrances may not be identical due to discrepancies arising during the cutting stage. Did the authors do an experiment where they ran a mixture of two amino-acids through the CNT channel and show that it exhibits the same type of discriminating power? That control experiment (which should probably be considered mandatory) will eliminate the possibility of discrimination arising from the entrance differences. How many nanotubes did the authors test and how many of them exhibited such strong discriminating behavior?

10. Fig. S12-14 need to show the same analysis that was used to build the Figure 4, that's the only way that a reader would be able to check for consistency of the results. The authors state that they have conducted these experiments "more than three times", but only show two results for each pair. Perhaps, there is a way to summarize the results for the rest of the measurements.

Reply to Reviewer 1

Questions & Answers:

Reviewer #1 (Remarks to the Author):

The report by Peng et al is the best quality CNT-based translocation sensing data out there and is based on adding an Density Gradient Ultra-centrifugation(DGU) purification step that has worked well for CNT gradation by size/chirality (Hersam group Northwestern Univ. ref 34). However, I hesitate to recommend it for publication in a top tier journal. The major detractors of the work are:

The main advance was to add a purification step to get the most uniform CNTs possible for the translocation experiments. However there was very little characterization of the CNTs outside of Raman, size exclusion series, and the conduction experiments themselves (inference only of CNT L&D). TEM characterization is desperately needed. The presence of RBM is encouraging, but Fig S3 c is significantly higher signal than b, and RBM is higher than G. So a few inconsistencies. Most of the data shown appears to come from one or a few different samples. If the primary advance of this work is a more uniform pore made possible by DGU, then a statistically valid number of different samples are needed.

Answer: Thanks for the valuable comments. We have conducted additional experiments to present a better characterization of the CNTs (Figure S1 & 4-6). Because the diameters of the CNTs are close to the limit of the TEM resolution, we used AFM to statistically analyze the diameter and length of the DGU-purified ultrashort CNTs.

Also, we have re-run the Raman spectra to fix the inconsistency (Figure S7). It is very likely that the presence of detergents on the surface of the CNTs severely affected the RBM peak in the Raman spectrum.

During the course of this work, we have cut and purify more than 20 different batches of CNTs. We did not notice any significant differences between batches.

They quote slightly enhanced mobilities (2-3), but conductance has a D^2/L dependence. These values have a high degree of uncertainty, thus it is not appropriate to report them without uncertainty calculated. I suspect the uncertainty will be on the order of the calculated value, not really appropriate for a top journal.

Answer: Thanks for the point. It is true that the enhanced ion mobilities could only be

estimated in a range due to the uncertainty of the length of the inserted CNTs. However, we have worked out the range of the ion mobilities in CNTs based on the experimental results (page 8, paragraph1). Besides, we further conducted the optical imaging of Ca^{2+} flux through CNTs to provide another evidence for the enhanced ion mobilities. Actually, the two groups of data matched quite well (Figures 2 and 3).

Fig S4 has quoted diameters of 1.7nm and 1.9nm for two different samples. Manuscript shows 1.2nm. The primary hypothesis of uniformity is not supported.

Answer: We apologize for the confusion that has been caused. We actually measured the diameters of different fractions in DGU separation, but the samples were not clearly labelled in the previous submission. Now we have labelled the samples in revised Figure S7.

The title has ‘Atomic resolution sensing.’ however this is not at all proven. The diameter control is not demonstrated to the angstrom scale precision. The molecules sensed (amino acids) differ by more than a few atoms. Asp vs Glu has an extra CH_2 link, giving more conformational flexibility. ‘Molecular sensing...’ would be more appropriate.

Answer: Thanks for the suggestion. We have changed the title to “Molecular sensing in ultrashort single-walled carbon nanotube nanopores”.

Fig 4 is the most compelling one, showing a distinction between amino acid monomers. It is not clear if the same trend would be repeated with different CNT diameter (i.e. different samples). Short peptide chains (~7-10 units) of programmed sequence can be easily purchased. This is the more appropriate target for sensing.

Answer: Thanks for the question. Actually, the amino acid discrimination could only be achieved with small-diameter CNTs from DGU fraction 1. During the revision, we also tested the translocation of short peptides through CNTs. We found that the 4-amino acid and 7-amnio acid peptides cannot pass through the CNTs in fraction 1, but it is likely they can translocate larger diameter CNTs in fraction 4. These results are listed below (**Figure R1-4**). Although some of them are quite interesting, they do not provide relevant information for peptide sequencing. We feel those results do not fit in the context of this work and might be saved for future studies.

Minor points: The abstracts first few sentences are almost contradictory. Are CNTs near perfect or highly inhomogeneous? Make more clear what the defects are. The

introduction reads as a competition between groups, best to leave group names out and focus on the advancement of the technology in the field. Ref. 20 should be explicitly cited as the first group to purify SWCNTs via DGU. Presently it is buried referring to supplemental figures.

Answer: Thanks for bringing up these points. Actually, there is no contradiction between “perfect” and “inhomogeneous”. The CNTs have “perfectly” round structure, but their diameter, chirality and length are “inhomogeneous”.

We have removed the group names in the introduction part and focused on the science itself.

We have moved Ref. 20 to a more conspicuous place and cited it as a major DGU separation reference.

Figure R1. Current traces and statistical analysis of the current events generated by the collision of polypeptide FGDDD with a SWCNT nanopore. (a) Typical current traces of the collision of polypeptide FGDDD with a SWCNT nanopore at cis side. The potential was held at +70 mV and current was 66 pA (conductance $G = 0.94$ nS). (b) Scatter plots of the current blockade I/I_0 versus event durations of the trace in a. (c) Histograms of the current blockades of polypeptide FGDDD collision with SWCNT. Red solid lines in the current blockade histograms are gaussian fit to the histograms. (d) Histograms of the dwell time of polypeptide FGDDD collision with SWCNT. Red solid lines in the dwell time histograms are single exponential fit to the histograms. (e) Curve of dwell time versus applied voltage for polypeptide FGDDD collision with SWCNT

nanopore. Data were acquired in the buffer of 1.0 M KCl, 10 mM Tris, pH 8.0 and 200 nM polypeptide FGDDD. The conductance of the SWCNT nanopore is 0.94 nS (from fraction 1).

Figure R2. Current traces and statistical analysis of the current events generated by the translocation of polypeptide FGDDD through a SWCNT nanopore. (a) Typical current traces of the translocation of polypeptide FGDDD through a SWCNT nanopore at cis side. The potential was held at +50 mV and current was 112 pA (conductance $G = 2.24$ nS). (b) Scatter plots of the current blockade I/I_0 versus event durations of the trace in a. (c) Histograms of the current blockades of polypeptide FGDDD translocation through SWCNT. Red solid lines in the current blockade histograms are gaussian fit to the histograms. (d) Histograms of the dwell time of polypeptide FGDDD translocation through SWCNT. Red solid lines in the dwell time histograms are single exponential fit to the histograms. (e) Curve of dwell time versus applied voltage for polypeptide FGDDD translocation through SWCNT nanopore. Data were acquired in the buffer of 1.0

M KCl, 10 mM Tris, pH 8.0 and 200 nM polypeptide FGDDD. The conductance of the SWCNT nanopore is 2.24 nS (from fraction 3).

Figure R3. Current traces and statistical analysis of the current events generated by the collision of polypeptide FGGDDDDDD with a SWCNT nanopore. (a) Typical current traces of the collision of polypeptide FGGDDDDDD with a SWCNT nanopore at cis side. The potential was held at +50 mV and current was 42 pA (conductance $G = 0.84$ nS). **(b)** Scatter plots of the current blockade I/I_0 versus event durations of the trace in a. **(c)** Histograms of the current blockades of polypeptide FGGDDDDDD collision with SWCNT. Red solid lines in the current blockade histograms are gaussian fit to the histograms. **(d)** Histograms of the dwell time of polypeptide FGGDDDDDD collision with SWCNT. Red solid lines in the dwell time histograms are single exponential fit to the histograms. **(e)** Curve of dwell time versus applied voltage for polypeptide

FGGDDDDDD collision with SWCNT nanopore. Data were acquired in the buffer of 1.0 M KCl, 10 mM Tris, pH 8.0 and 200 nM polypeptide FGGDDDDDD. The conductance of the SWCNT nanopore is 0.84 nS (from fraction 1).

Figure R4. Current traces and statistical analysis of the current events generated by the translocation of polypeptide FGGDDDDDD through a SWCNT nanopore. (a) Typical current traces of the translocation of polypeptide FGGDDDDDD through a SWCNT nanopore at cis side. The potential was held at +50 mV and current was 315 pA (conductance $G = 6.31$ nS). **(b)** Scatter plots of the current blockade I/I_0 versus event durations of the trace in **a**. **(c)** Histograms of the current blockades of polypeptide FGGDDDDDD translocation through SWCNT. Red solid lines in the current blockade histograms are gaussian fit to the histograms. **(d)** Histograms of the dwell time of polypeptide FGGDDDDDD translocation through SWCNT. Red solid lines in the dwell time histograms are single exponential fit to the histograms. **(e)** Curve of dwell time

versus applied voltage for polypeptide FGGDDDDDD translocation through SWCNT nanopore. Data were acquired in the buffer of 1.0 M KCl, 10 mM Tris, pH 8.0 and 200 nM polypeptide FGGDDDDDD. The conductance of the SWCNT nanopore is 6.31 nS (from fraction 4).

Reply to Reviewer 2

Questions & Answers:

Reviewer #2 (Remarks to the Author):

The submission by Peng et al describes the preparation of single-walled carbon nanotubes (SCWNTs), their in-depth characterization in terms of unique conductance properties, and applications for single-amino acid discrimination which is of relevance for label-free sensing applications.

The preparation involved density gradient ultracentrifugation to obtain up to several nm-short tubes of narrow width distribution in the range of 1 nm. This has so far not been used and obtained.

The chemical nature of the pores was analyzed with Raman spectroscopy, their width with probe molecules of various sizes in electrical recordings, and their ion selectivity and pH-dependent ion conductivity to confirm the influence of diameter and terminal COOH groups. Using electrical recordings and optical imaging of transport, the pores were found to show unusually high conductance which was attributed to the high mobility of electrolyte ions in the SCWNTs.

The high conductance of the tubes was applied to distinguish amino acid pairs. The clear distinction between structurally similar leucine vs isoleucine residues is striking.

The manuscript could be of interest for Nature Communications given the newly applied approach to obtain defined SCWNTs, the thorough characterization of the tubes (exception see below), and the exploitation of the unique conductance properties for a previously very difficult to achieved discrimination of analytes via nanopore sensing. The manuscript is clearly structured and generally well written.

Before the manuscript can be considered for publication, the following points should be addressed.

Answer: We sincerely appreciate your favorable comments.

Major:

Length characterization: The authors claim a length of 5-10 nm for the carbon tubes. But the length of the tubes was not experimentally determined. This should be done such as via AFM.

Answer: Thanks for the suggestion. Now we have added length characterization of CNTs in Figure S4 and S5.

The experimentally determined higher conductance of carbon nanotubes should be quantitatively compared with existing data on other, longer carbon nanotube conductance. The higher conductance should be explained by referring and updating biophysical models or potentially simulations which have been developed previously.

Answer: The only ionic conductance values through single long CNTs were reported by Lindsay and Nuckolls in *Science*, **2010**, 327, 64. The conductance range in their work spans 4 orders of magnitude and many of them exceeds theoretical values (metallic tubes). In this work, we separated ultrashort CNTs by DGU and narrowed down the conductance range to 0.1-10 nS.

The higher conductance in short CNTs was proposed by Hummer *et al.* in a molecular dynamic simulation study (*Biophys. J.* **2005**, 89, 2222–2234). Their conclusion proposed an ~50% increase in ion mobilities inside short SWCNTs, which is in qualitative agreement with our results. There is another simulation paper dealing with ultrafast ion transport in CNTs (Phys. Fluids 2017, 29, 092003), however, the enhanced conductance was only observed in CNTs that are several hundred nanometers long.

The units for equation 1 are strange

G should be S or kg⁻¹ m⁻² s³ A²

D²(uK)c(KCl) e is m² V⁻¹ s⁻¹ mol L⁻¹ m² m⁻¹ or m³ kg⁻¹ m⁻² s³ A mol L⁻¹

Giving

kg⁻¹ m⁻² s³ A² = m³ kg⁻¹ m⁻² s³ A mol L⁻¹

Giving a proportionality factor with units of

m⁻³ mol⁻¹ L A

Answer: We have re-checked the units for equation 1. The two sides actually match.

$$\Delta G = 6.02 \times 10^{26} \times \frac{1}{L_{CNT}} \left(-\frac{\pi}{4} d_{ITP}^2 (\mu_K + \mu_{Cl}) c_{KCl} e + \mu_K^2 q_{ITP}^2 \right)$$

$$\Delta G \sim \frac{A^2 \cdot s^3}{kg \cdot m^2}$$

$$\frac{1}{L_{CNT}} \sim m^{-1}$$

$$d_{dTTP}^2 \sim m^2$$

$$(\mu_K + \mu_{Cl}) = \frac{v_K + v_{Cl}}{E} = \frac{v_K + v_{Cl}}{U/L} = \frac{v_K + v_{Cl}}{IR/L} \sim \frac{m/s}{A \cdot \frac{kg \cdot m^2}{A^2 \cdot s^2} \cdot m^{-1}} = \frac{s^2 \cdot A}{kg}$$

$$c_{KCl} = \frac{n_{KCl}}{V} = \frac{n_{KCl}}{L} = \frac{10^3 \cdot n_{KCl}}{m^3} \sim m^{-3}$$

$$e = 1.602 \times 10^{-19} C \sim C = A \cdot s$$

$$Right \sim \frac{1}{L_{CNT}} \left(-\frac{\pi}{4} d_{dTTP}^2 (\mu_K + \mu_{Cl}) c_{KCl} e + \mu_K^2 d_{dTTP}^2 \right) \sim m^{-1} \times m^2 \times \frac{s^2 \cdot A}{kg} \times m^{-3} \times A \cdot s = \frac{A^2 \cdot s^3}{kg \cdot m^2} \sim \Delta G$$

Minor points:

Page 3: The historic account is one way of introducing the topic but scientifically likely not the most exciting. It could be better to restructure the first introduction paragraph to follow scientific concepts and ideas.

Answer: Thanks for the suggestion. We have re-written the first paragraph to focus on the development of scientific concepts.

Page 6, line 114: Should the pH-dependence not follow a sigmoidal dependence similar to typical titration experiments?

Answer: From the conductance-pH relationship in Figure R5, it seems the conductance only changes monotonically upon pH tuning within 3.0 to 8.0.

Figure R5. Conductance-pH relationship of three SWCNTs.

Page 7, line 132: Define the distinct sensing mechanism.

Answer: The sensing mechanism has been explained in the text.

Page 9: Why was the blinking observed? Likely as dTTP blocked passage of the Ca^{2+} ion. This should be explained.

Answer: Thanks for the suggestion. It is true that dTTP blocked the passage of the Ca^{2+} ions and led to the observed fluorescent blinking. The explanation has been added in the text (page 9).

Page 10, line 193: The authors ignore the recent break-through work by Dekker et al in Science on nanopore-mediated peptide sequencing. This paper should be cited. The authors' motivation into the amino acid sensing via their carbon nanotube has to be edited accordingly.

Answer: Thanks for the reminder. We have cited Prof. Dekker's breakthrough work on nanopore peptide sequencing in the revision. Accordingly, we rephrased the motivation to

conduct peptide sequencing with CNT nanopores.

Page 11, line 225: Distinguishing amino acids via multiple carbon nanotubes of different selectivities can in my view only be applied for peptide sequencing when the nanopores are in a serial array. But this has so far not been achieved. The authors likely refer to a parallel array of different nanopores and amino acid selectivities which is very difficult to perceive to achieve sequencing. The paragraph should be edited to spell out more clearly what and what cannot be achieved with their pores in terms of peptide sequencing.

Answer: Thanks for bringing up the point. From many groups of results we had obtained, we found that although many of the SWCNTs in DGU fraction 1 could discriminate the 3 pairs of amino acids in Figure 4, some are better than others in distinguishing a specific pair of amino acids. To build a sensing array, we need to find the optimal SWCNT nanopore for each amino acid/pair and implement them in parallel. For each unknown sample, we need to run sensing experiments through all the nanopores in the array and then pick out the best match. But I feel it is a bit far-fetched to describe these details in this work because there is a lot more to do before that.

The conclusions chapter is weaker than the results section and should be brought to a higher level. It should feature a more expanded comparison to state of the art, a comparison of the pros and cons of carbon nanotubes for sensing, and more detailed plans on how to develop the technology further.

Answer: Thanks for the suggestion. We have re-written the conclusion part and made it stronger.

The formatting of the references is inconsistent.

Answer: This has been fixed.

Reply to Reviewer 3

Questions & Answers:

Reviewer #3 (Remarks to the Author):

Weichao Peng and coworkers isolated ~5-10 nm SWCNT and reconstituted into lipid bilayers. The nanotubes were capable of translocating(?) amino acids, and differences with similar amino acids were identified by current recordings. This is an important work, as the mechanism of molecular recognition in nanopores is unknown, and especially in SWCNT-nanopores. Here, the observation that molecules as small as individual amino acids can be distinguished is intriguing, as the inside of a CNT is chemically uniform. Hence, this work would provide important additional data to understand the recognition mechanism. Furthermore, the ability of distinguishing enantiomeric molecules with nanopores will most likely have important applications.

Answer: Thanks for the positive comments.

It will be important, however, to prove whether or not the analytes enter the CNT. The authors should, therefore, provide a voltage dependence of the dwell times for the molecules analyzed. If the dwell time decreases with the bias, then this can be taken as a proof of molecular translocation.

Answer: Thanks for the suggestion. The voltage dependence experiments have been added in Figure S11.

Another point concerns the ability of these CNT to distinguish between similar molecules. As far as I understood, the authors added one analyte on one side of the CNT and another on the other side, and they observe a difference between the two molecules. Why? Could this asymmetry be explained by the fact that the two molecules interact with the entry of the CNT rather than translocating through it. If they have not done so, the authors should add both analytes on the same side and observe the two distributions.

Answer: Thanks for the point. The both ends of CNTs are supposed to be the same. Now, we have run the amino acid discrimination experiments on the same side and observed two distinct distributions. The new data are added in Figure S16-18.

Other points

Line 46. What is a “homemade device”?

Answer: This was quoted from the original paper. It is a device in which one SWCNT spans a barrier between two fluid reservoirs.

Line 51-52. What is the difference between current blockades and spikes?

Answer: Thanks for the question. Current blockades mean the current decreases in the presence of DNA molecules, whereas current-increasing spikes mean the current actually increases when DNA is translocated through the SWCNT.

Line 54. Please indicate what are the difficulties on getting the SWCNT.

Answer: We rephrased to “the difficulties of lacking a methodology for obtaining homogeneous SWCNTs”.

Line 80. The connection with protein sequencing is not obvious.

Answer: We rephrased the sentence “...similar targets such as amino acids in SWCNT nanopores...” to make the connection more obvious.

Line 82. Please define what is DGU separation and what is the advance compared to previous protocols used.

The authors should discuss why they have such a large variability in the range 0 - 2 nm. It should be noted that, compared to biological nanopores, this distribution is actually rather wide. Is this due to the different length of the CNT? Or different diameters? Further it appears that there are several distributions (0.2 – 0.4 – 0.7 – 0.9)

Answer: The DGU is defined in line 72 where it first appeared. Compared with previous protocol which is based on HPLC separation, DGU separation has the advantage of obtaining SWCNTs with narrow range diameter.

There are several reasons for the conductance of CNTs to have a wide range. Both length and diameter have significant influence on the conductance of CNTs. But other parameters such as chirality, semi-conducting or metallic properties, also have effects on the conductance.

The distribution is totally random. We have added new conductance data in Figure 1b, and it looks better now.

Line 89. Where does the equation come from (reference)? The equation should be re-drawn, as it is also not easily understandable.

Answer: We added the reference at the end of the sentence. The equation has been re-drawn in the revised version.

Line 96. “Literature method” please rephrase

Answer: We have rephrased it to “a method reported in literature”.

Line 98. The author mention they use the CNT in the range 0.8-1 nS. However, in Fig. 2C and Fig 4 the SWCNT the distribution appears larger.

Answer: The conductance of SWCNTs we used in Figure 2c and 4 is still in the range of 0.8-1.0 nS.

Line 99. “The results demonstrate...”. Which results? Not clear to me what are the authors referring to.

Answer: We have rephrased the sentence to “The results in Fig. 1b, 1c and Supplementary Fig. 7 demonstrate...”.

Figure 1e. What do Tube 1-2-3 refer to? Three different CNT? If so, why there are error bars?

Answer: Tubes 1, 2, 3 refer to three different CNTs. Because the pH tuning experiments was conducted in-situ, we recorded 3 traces at different time to obtain the average values.

Figure S5. Why there are no error bars? How many CNT were tested at each pH?

Answer: The error bars are added. We have tested more than 20 different CNTs. The results are quite similar.

Line 105. Please indicate where to find the IV curves.

Answer: The I-V curves are in Figure S8.

Line 112. Please indicate the ion selectivity of the 0.8 nm CNT

Answer: The ion selectivity of the 0.8 nm CNT is around 40.

Figure 2c. The filtering rate is 5kHz. Why? Can the author use higher filtering? In addition, the authors should add a better representation of individual blockades (more blockades and a better expansion). Why in figure 2c the noise of the experiment increases after adding the sample? Finally, the authors should indicate sampling and filtering in the legend.

Answer: We have tested different filtering rate 10 kHz, 5 kHz and 2 kHz for recording current traces. We found that 10 kHz and 5 kHz filtering give similar current traces but 10 kHz filtering has a larger noise baseline.

More individual blockades have been added in Figure 2c.

The increased noise after the addition of the sample might result from the CNT-dTTP colliding events. Those short and dense spikes look like noise in a large timescale.

We have added the sampling rate and filtering rate in the legend.

Line 122. Is there any reason why the authors used dTTP? In addition, the authors should perform a voltage dependency to check whether the molecules translocate across the nanopore. This is important, because if the molecule do not translocate it will reveal an important information about the mechanism of molecular identification.

Answer: Thanks for the question. Because the diameters of the SWCNTs we used in this work are too small to allow DNA translocation, we then tested the four nucleotides (dNTP). Among the four nucleotides, dTTP affords distinct and uniform translocation events. Therefore, we chose dTTP as the model molecule.

We have performed the voltage-dependent dTTP translocation experiments and the

results are added in Figure S11. It is found that the dwell time of dTTP translocation (τ_{off}) exhibits a maximum value versus applied voltage, which is generally considered as evidence for translocation.

Line 127. The authors say: “Other than that, the current blockages caused by the presence of dTTP...” what do they mean? Do they refer to the Ires? How was defined?

Answer: The current blockage in this work is defined as I/I_0 , where I is the current amplitude blocked by the analyte and I_0 is the open pore current.

Line 131. Is the capture rate higher because the events are longer (missing events)?

Answer: The higher capture rate probably results from the stronger interactions between the inner wall of SWCNTs and dTTP molecules.

Line 135. “... bulk mobility”. Please add a reference.

Answer: We have added two references to this sentence (Refs. 25 and 26).

Lines 144-147. Please indicate the ionic strength and applied bias for all the conductance.

Answer: This information is in the Figure 2 legend.

Line 147. The range 5-10 nm for the CNT is rather large. Would this affect the ability of the CNT to detect molecules? How can the author distinguish between length and diameter of the CNT if they only select based on the conductance of the CNT?

Answer: This is the bottleneck of the project. Currently, there is no way for us to simultaneously determine the CNT length and diameter. However, from more than 100 sensing experiments, we could conclude that the abilities of the CNTs to detect molecules are not affected by this predicament. The results of molecular sensing are highly reproducible for each fraction of the CNTs.

Line 197. Is sequencing by enzymatic cleavage proposed in this work? If it is, the authors should elaborate more (e.g. how can this be done?). Otherwise, they should add a

reference (and perhaps still elaborate). It is not obvious how measuring amino acids can lead to protein sequencing.

Answer: Thanks for bring up this point. Indeed, we propose to sequence a peptide through enzymatic cleavage followed with discrimination of amino acids. Measuring amino acids is only one step in this process. We elaborate the whole strategy in the conclusion section as suggested by another Reviewer.

Line 209. Please add all the analytes in the same chamber.

Answer: We have run these experiments and added the new data in Figure S16-18.

Line 2017. What is a ‘skeletal’ structure?

Answer: The skeletal structure of an organic compound is the series of atoms bonded together that form the essential structure of the compound.

Line 2018-2019. Please explain the “difference” in the translocation profiles

Answer: The main differences are current blockage and capture rate. This has been added in the text.

Reply to Reviewer 4

Questions & Answers:

Reviewer #4 (Remarks to the Author):

Peng, Liu, Wu and co-workers report ion transport and small molecule blockade measurements in short carbon nanotubes of ~1.2 nm diameter inserted into a lipid bilayer matrix. Ion conductance measurements show that ion electrophoretic mobility shows modest enhancement of 2-4x in these pores. The authors also report combined optical and ion transport experiments showing Ca²⁺ ion translocation in CNT pores using a droplet interface bilayer setup. Finally, the authors report that CNT pores can discriminate between several pairs of amino-acids that are often hard to discriminate with remarkable efficiency. Even though the results, and particularly the demonstration of Ca²⁺ ion transport and the amino-acid pair discrimination detection is impressive, this paper raises a number of questions that do not allow me to endorse it, at least in the current form.

Specifically, I find the central result of the unusually strong discrimination of the three pairs of amino-acids by the short carbon nanotube channels interesting (although that part lacks a critical control experiment). I find the mechanistic explanations offered by the authors much less convincing. The other part of the manuscript that reports slightly enhanced electrophoretic mobility of the salt ions in the channel is also not convincing and relies on questionable analysis. The Ca²⁺ ion transport measurements are elegant and fascinating, but they do not seem to connect to the rest of the manuscript narrative and results (which were taken using a different experimental setup). Perhaps refocusing the manuscript on a coherent linear narrative around amino-acid pair discrimination, adding critical controls, and presenting more robust statistics would improve the impact of the work. Some specific comments follow:

Answer: Thanks for the insightful comments. We have run additional experiments and revised the manuscript to address all the questions.

1. The connection between "special feature" of the nanotube pores and the enhanced sensing sensitivity discussed on the page 4 is questionable. In fact, this theory of the authors goes more or less against the conventional wisdom of the nanopore field. Interactions of the sensing molecule with the nanopore wall affect the duration of the blockade, but not the blockade amount, which is largely determined by excluded volume of the molecule. From that point of view a CNT will have a strong disadvantage as a nanopore discriminator over the more conventional pores such as MspA or α -haemolysin

(a-HL) that have a defined constriction in the channel that makes for an effectively shorter sensing region. No argument is provided for why authors' conclusion should be correct. In fact it is likely wrong.

Answer: Thanks for the point. We have pointed out in the text that CNT might adopt a different sensing mechanism from that of MspA or alpha-hemolysin. The protein pores all have a constriction/short sensing region for recognizing the targets, but the uniform interior of CNT could interact with the target molecules via non-covalent interactions such as π - π , CH- π , or hydrophobic interactions. The overall sensing performance will depend on what the target molecules are and what kind of interactions between nanopore and targets.

“Interactions of the sensing molecule with the nanopore wall affect the duration of the blockade, but not the blockade amount, which is largely determined by excluded volume of the molecule.” This statement is true when the ion mobilities are all the same. The equation 1 about conductance change we gave in the text explicitly involves ion mobilities. Therefore, if the ion mobility is enhanced, the blockage amplitude will be increased.

2. The same discussion on page 4 also states that faster ion transport in CNTs would lead to higher signal and higher resolution in small molecule translocation studies. It is hard to see how one follows from the other. Even if ions flow faster in the CNT pore, the small molecules would also flow faster, which would only result in shorter duration blockades, unless the authors provide a mechanism for why those small molecules are slowed down.

Answer: There is a misunderstanding of the sensing mechanism here. We observe the current blockage caused by the small molecules because these molecules interact with the nanopore and consequently block the ionic current through nanopore. If ions flow faster, the interactions between the molecule and the nanopore will not change. Therefore, the current blockade duration will NOT change. But the ionic current change which is blocked by the molecule will be increased due to the higher ion mobility.

3. The manuscript claims a narrow conductance range for the short CNTs that they synthesized. Figure 2B shows a rather wide range (and rather sparse statistics with many histogram bars at 1 even count). An average conductance number and an error bar would be helpful in evaluating the quality of the manuscript's statement.

Answer: We claimed a narrow conductance range of CNTs in this work because the conductance of the same types of CNTs can span 4 orders of magnitude (*Nat. Commun.*

2013, 4, 2989). We have added new data in Figure 1b during revision. It is true that the conductance range of CNTs is still quite wide. But for most of the experiments we have run in this work, we chose to use the CNTs with conductance range of 0.8-1.0 nS and this has greatly improved the homogeneity of the CNTs.

4. What was the initial range of diameters for the long CNT stock? This information may be helpful in evaluating the quality of the purification process.

Answer: The initial range of diameters of the long CNTs is 1.2-1.7 nm (on the labels). The long CNTs were purchased from Nanointegris.

5. The data on the Fig 2 a-c need to be presented on the same scale to make a fair comparison. The data also show that the absolute level of current blockade by dTTP is roughly similar between the CNT and MspA (about 20 pA), but is much smaller for the case of α -HL (roughly 5 pA). Intuitively, you would expect a similar absolute current blockade value, as it is related to an excluded volume of the molecule being transported. Can the authors explain this discrepancy?

Answer: Thanks for the suggestion. We have adjusted the scale of Fig. 2a-c to make a better comparison. Actually, the current blockage by dTTP in CNT is about 20 pA and those in MspA and alpha-hemolysin are around 5 pA. The explanation for this phenomenon would be the same for question 2. The excluded volume of the molecule is the same, but the current blockade value by the molecule could be enhanced in SWCNT nanopore owing to the faster ion transport.

6. The use of equation 1 on page 7 to interpret the experiments on dTTP is incorrect. Even though dTTP is a relatively large molecule, its size is still much smaller than the length of the pore. The model of Smeets and Dekker (Eq.5 in their paper is nearly identical to Eq. 1) was derived for translocating long DNA through the channel where DNA occupies the whole channel length, which is quite different from a case of small molecule translocating through a longer channel. How do the authors justify using the same $1/L$ factor in their equation?

Answer: Thanks for the question. Equation 1 deals with two competing effects when an analyte is translocated through a nanopore. On one hand, the conductance is decreased because of the volume that is occupied by the analyte. On the other hand, the counterions shielding the charge of the analyte add a positive contribution to the ionic current. Therefore, there is no such a precondition that the analyte should occupy the whole

channel length. In a recent publication, the conductance changes of nanobeads passing through a micropore were also dealt with using this model (*ACS Sens.* **2020**, *5*, 3892–3901).

7. The comparison of conductances at the bottom of page 7 borders on disingenuous. Even if we ignore that MspA and aHL pores are not cylindrical and have a pronouncedly conical shape, it looks like the authors are arbitrarily assuming the length of their channels to be the lower boundary of their length estimates (5 nm) to obtain a better agreement with their enhancement factors. What are the errors associated with the measurements and do they even allow for a solid claim of the enhanced mobility in these channels?

Answer: We clearly used $L_{\text{SWCNT}} \approx 5.0\text{-}10.0$ nm rather than 5 nm when we estimated the enhanced ion mobility in SWCNT. Therefore, our conclusion is “...ion mobilities μ_{K} or μ_{Cl} in SWCNT nanopores are **2-4 times** higher than that in biological nanopores...”.

The conductance measurement experiments involve both K^+ and Cl^- . But the optical imaging experiments allow us to focus on the mobility of the cations. This also helps us corroborate the conclusion we obtained in the conductance measurements.

8. Figure 3C needs to show a zoomed-in section to show a correlation between the optical signal and ionic current images to make it easier to see the correlation. I had to blow it up and use a ruler.

Answer: The traces shown in Figure 3b and 3c were recorded on the same DHB device but were not simultaneously recorded.

9. Fig. 4 data: Even though CNT channels are symmetric inside, the two entrances may not be identical due to discrepancies arising during the cutting stage. Did the authors do an experiment where they ran a mixture of two amino-acids through the CNT channel and show that it exhibits the same type of discriminating power? That control experiment (which should probably be considered mandatory) will eliminate the possibility of discrimination arising from the entrance differences. How many nanotubes did the authors test and how many of them exhibited such strong discriminating behavior?

Answer: We have now added the new results of the individual amino acid sensing experiments together with their mixture sensing experiments in Figure S16-18. The results show no sign that the entrance plays any role in the discrimination of amino acids.

For the discrimination of amino acids in Figure 4, we have carried out about 60 experiments. Among 17 attempts for Asp and Glu, 14 runs are successful (success rate 82%); among 22 attempts for Phe and Tyr, 17 runs are successful (success rate 77%); among 20 attempts for Leu and Ile, 12 runs are successful (success rate 60%).

10. Fig. S12-14 need to show the same analysis that was used to build the Figure 4, that's the only way that a reader would be able to check for consistency of the results. The authors state that they have conducted these experiments "more than three times", but only show two results for each pair. Perhaps, there is a way to summarize the results for the rest of the measurements.

Answer: We have revised the figures according to the suggestion.

Reviewers' Comments:

Reviewer #1:

Remarks to the Author:

I thank the authors for being generally responsive to the comments and performing important new experiments. In particular the TEM to show CNTs and AFM based lengths. In page 8 para 1, they quote 2-4 times bulk mobility. They should still report the observed mobilities values for given lengths (assuming a CNT length of 5 and 10nm). Better yet would be to use the histogram data of figure S5a. To my eye looks like 10+/- 5 nm. I think that works out 3-5 x bulk mobility. Ref 28 saw similar enhancements (~3) so should include that reference with the others as being consistent with your observation. The Ca²⁺ optical imaging isn't a good measure of mobility, but conductance is sufficient to me. Other comments seem addressed to me. I do think the term on page 1 of "homemade" should be removed even though the other group improperly used it. Certainly not made at someone's house. None of these can be purchased so fabricated in a lab. I recommend publication since this is the best example of CNT diameter control for CNT channels.

Reviewer #2:

Remarks to the Author:

The points raised in the first round of reviewing have been successfully settled. The manuscript is now publishable in Nature Communications.

Reviewer #3:

Remarks to the Author:

The authors have addressed all my previous comments. I now recommend the manuscript for publication in Nature Communications

Reviewer #4:

Remarks to the Author:

I appreciate the additional work done by the authors and the new data added to the manuscript, which increase my confidence in the experimental results. I still have a number of serious question about the data interpretation and still some about the data presented.

The added AFM data (Fig. S3) are substandard as only three CNTs are analyzed, The AFM images show significant height variation along the CNTs, is that a surfactant use to disperse CNTs? The histogram of the CNT diameters shown on the Figure S4 is rather wide. I suspect that the actual diameter distribution is more narrow, thus it is unfortunate that the authors do not have access to HR-TEM to have a better look into the CNT diameters.

Question 1: I find the authors' answer unconvincing. I still do not understand how an increased ion mobility in the nanotube will affect the ability of that nanotube to discriminate between two molecules, as those molecules go through the same channel and experience the same increased mobility. I still do not think any evidence of a different sensing mechanism for transport in CNTs relative to sensing in regular protein nanopores.

Question 2: I again find the authors' answer unconvincing.

Question 5. Again, when the molecule is shorter than the channel, faster transport should lead to a shorter blockade, not to the blockade value enhancement.

Question 6: The answer, unfortunately, does not address the substance of the question. An improper use of the model in another article does not prove that the model is applicable.

Question 8: If the traces were not recorded simultaneously, what does the Figure 3C tell us? The data are drawn to suggest that the zoomed-in parts correspond to the same events, while in reality they are not representing the same events?

Reviewer #1 (Remarks to the Author):

I thank the authors for being generally responsive to the comments and performing important new experiments. In particular the TEM to show CNTs and AFM based lengths. In page 8 para 1, they quote 2-4 times bulk mobility. They should still report the observed mobilities values for given lengths (assuming a CNT length of 5 and 10nm). Better yet would be to use the histogram data of figure S5a. To my eye looks like 10+/- 5 nm. I think that works out 3-5 x bulk mobility. Ref 28 saw similar enhancements (~3) so should include that reference with the others as being consistent with your observation. The Ca²⁺ optical imaging isn't a good measure of mobility, but conductance is sufficient to me. Other comments seem addressed to me. I do think the term on page 1 of "homemade" should be removed event though the other group improperly used it. Certainly not made at someone's house. None of these can be purchased so fabricated in a lab. I recommend publication since this is the best example of CNT diameter control for CNT channels.

Answer: Thanks for the valuable suggestion and the positive comments. We have revised the ion mobilities in CNTs to be 3-5 times bulk mobility and included Ref 28 accordingly.

Page 8, line 12-14. "This agrees well with our previous study of DNA translocation in larger-diameter SWCNT nanopores⁶ and a study of ion mobilities in SWCNT membranes²⁸."

The term "homemade" has been deleted.

Reviewer #2 (Remarks to the Author):

The points raised in the first round of reviewing have been successfully settled. The manuscript is now publishable in Nature Communications.

Answer: Thanks for the questions and the support.

Reviewer #3 (Remarks to the Author):

The authors have addressed all my previous comments. I now recommend the manuscript for publication in Nature Communications.

Answer: Thanks for the questions and the support.

Reviewer #4 (Remarks to the Author):

I appreciate the additional work done by the authors and the new data added to the manuscript, which increase my confidence in the experimental results. I still have a number of serious question about the data interpretation and still some about the data presented.

The added AFM data (Fig. S4) are substandard as only three CNTs are analyzed, The AFM images show significant height variation along the CNTs, is that a surfactant use to disperse CNTs? The histogram of the CNT diameters shown on the Figure S4 is rather wide. I suspect that the actual diameter distribution is more narrow, thus it is unfortunate that the authors do not have access to HR-TEM to have a better look into the CNT diameters.

Answer: Thanks for the points. Now we have added the AFM data of all the 10 fractions in Figure S5. We used surfactants (sodium cholate and sodium dodecyl sulfate) for dispersing CNTs, and it is likely that some surfactant aggregates remained in the sample solution for the AFM characterization.

We do have access to HR-TEM in our institute but the chief scientist who is running the HR-TEM told us that the diameter and length of our CNTs are not suitable for HR-TEM. Even if HR images of short CNTs could be taken, they are not reliable for the statistical analysis of the diameter.

Question 1: I find the authors' answer unconvincing. I still do not understand how an increased ion mobility in the nanotube will affect the ability of that nanotube to discriminate between two molecules, as those molecules go through the same channel and experience the same increased mobility. I still do not think any evidence of a different sensing mechanism for transport in CNTs relative to sensing in regular protein nanopores.

Question 2: I again find the authors' answer unconvincing.

Question 5. Again, when the molecule is shorter than the channel, faster transport should lead to a shorter blockade, not to the blockade value enhancement.

Answer: The above three questions all point to the same sensing mechanism issue, therefore they are answered together.

First of all, the sensing mechanism in CNTs is fundamentally the same as that in protein nanopores or any other types of nanopores. It is based on the detection of ionic current

blockades caused by the hindered movement of analytes. This hindered movement is due to the interactions between the analyte and the recognition site in nanopore. Without these interactions, the current would remain constant.

Second, the data of this work, together with previous studies from both our own and other groups, have established that the ion mobilities in CNTs are 3-5 times faster than the bulk mobility. Higher ion mobility results in higher recorded current signals. When an analyte is completely hindered inside a protein nanopore or a CNT nanopore, it blocks certain ionic current in both cases. However, because the ion mobility is 3-5 times faster in CNTs, the current blockade would be correspondingly higher (more ions are blocked across a surface per second, similar to the cases where one uses high concentration of electrolytes to enhance current resolution). This had been experimentally validated (*Nat. Commun.* **2013**, *4*, 2989).

Third, when two similar molecules are to be discriminated by a nanopore, the differences in their blocked currents would be the larger, the better. If we assume that a CNT and a protein pore possess the same size parameters, the differences in current blockades caused by two similar molecules would be a few times higher in a CNT nanopore than in a protein nanopore. It is the foundation of this work.

Question 6: The answer, unfortunately, does not address the substance of the question. An improper use of the model in another article does not prove that the model is applicable.

Answer: In the paper by Smeets and Dekker, there is no such prerequisite that the analyte should occupy the whole channel length. Generally, the length of protein pores is above 10 nm, and that of solid-state pores is even larger. If the equation should not be used when the analyte cannot occupy the whole channel, all the studies of short DNA (< 30 nt) translocation through nanopores in the literature should be reconsidered.

Question 8: If the traces were not recorded simultaneously, what does the Figure 3C tell us? The data are drawn to suggest that the zoomed-in parts correspond to the same events, while in reality they are not representing the same events?

Answer: We have recorded the traces with simultaneous electrical and optical recordings. Figure 3b and 3c are now replaced with the new data. The expanded views of the events correspond to the same events from electrical and optical recordings respectively.

Reviewers' Comments:

Reviewer #5:

Remarks to the Author:

This work by Peng et al. shows intriguing experimental evidence for the selective translocation of several amino acids through ultrashort (5-10 nm) single-walled carbon nanotube nanopores. One of the major technical challenges is preparing these nanopores with controlled diameters and lengths. The authors adopted multiple literature methods to address this challenge, and their data appear cleaner and reveal new insight previously unattainable with less controlled samples. The work is solid and novel, and except for a few minor issues, as noted below, the manuscript is well written. Publication is recommended.

Regarding equation 1, this model is briefly described in Cees Dekker's 2006 paper (ref. 27). Clearly, the geometrical term arises from ions conductance through a cylindrical pore with diameter D and length L . I agree with Reviewer 4 that for this equation to stand, the analyte, such as a DNA string, should be no smaller than L . My reasoning is as follows:

Consider a "rod-like" DNA string with an extended length of L_a and diameter D_a inside the nanotube pore of L .

In the case of $L_a \gg L$, the nanopore is reduced to a pore with the geometrical resistance dictated by cross-sectional $\pi/4 \cdot (D_a^2 - d^2)$ and L . This effectively leads to the equation the authors gave, considering the associated ionic effects and effective access to the pore.

For $L_a < L$, however, the nanopore is better treated as two different pores connected in a series:

pore 1: cross-sectional $\pi/4 \cdot (D_a^2 - d^2)$ and L_a

pore 2: cross-sectional $\pi/4 \cdot D^2$ and $L - L_a$

The total geometrical resistance of the pore is then given by $1/R = 1/R_1 + 1/R_2$. Therefore, the effect of L_a on the conductance change depends on the relative sizes of L_a and L .

Equation 1 is a rather simple model that does not consider many other phenomena that may occur for nanopores. See, for example, <https://pubs.acs.org/doi/10.1021/acs.jpcc.9b02178>. Nevertheless, it is still a useful model to include, if its limitations in the context of the current work can be discussed and clarified.

Minor technical issues:

How are the AFM samples prepared? What causes the noise pattern of the substrate shown in Figure S4b? Is the noise due to surfactants or other molecules used in the process?

It is difficult to see the ultrashort 5-10 nm nanotubes in Figure S5a. Is there a high-resolution AFM image to better show those ultrashort nanotubes? The nanotube length is critical to understand the data reported here.

The RBM peaks for some of the ultrashort nanotube fractions, such as Figure S6d (Fraction 2 and 4), are unusually sharp. Are they really Raman signals from the nanotube fractions? What is the excitation line?

Reviewer #5 (Remarks to the Author):

Reviewer #5 (Remarks to the Author):

This work by Peng et al. shows intriguing experimental evidence for the selective translocation of several amino acids through ultrashort (5-10 nm) single-walled carbon nanotube nanopores. One of the major technical challenges is preparing these nanopores with controlled diameters and lengths. The authors adopted multiple literature methods to address this challenge, and their data appear cleaner and reveal new insight previously unattainable with less controlled samples. The work is solid and novel, and except for a few minor issues, as noted below, the manuscript is well written. Publication is recommended.

Answer: Thanks for the positive comments and valuable suggestions.

Regarding equation 1, this model is briefly described in Cees Dekker's 2006 paper (ref. 27). Clearly, the geometrical term arises from ions conductance through a cylindrical pore with diameter D and length L . I agree with Reviewer 4 that for this equation to stand, the analyte, such as a DNA string, should be no smaller than L . My reasoning is as follows:

Consider a "rod-like" DNA string with an extended length of L_a and diameter D_a inside the nanotube pore of L .

In the case of $L_a \gg L$, the nanopore is reduced to a pore with the geometrical resistance dictated by cross-sectional $\pi/4 \cdot (D_a^2 - d^2)$ and L . This effectively leads to the equation the authors gave, considering the associated ionic effects and effective access to the pore.

For $L_a < L$, however, the nanopore is better treated as two different pores connected in a series:

pore 1: cross-sectional $\pi/4 \cdot (D_a^2 - d^2)$ and L_a

pore 2: cross-sectional $\pi/4 \cdot D^2$ and $L - L_a$

The total geometrical resistance of the pore is then given by $1/R = 1/R_1 + 1/R_2$. Therefore, the effect of L_a on the conductance change depends on the relative sizes of L_a and L .

Equation 1 is a rather simple model that does not consider many other phenomena that may occur for nanopores. See, for example, <https://pubs.acs.org/doi/10.1021/acs.jpcc.9b02178>. Nevertheless, it is still a useful model to include, if its limitations in the context of the current work can be discussed and clarified.

Answer: Thanks for the instructive discussions about this issue. We totally agree with the Reviewer that when $L_a < L$, the conductance change would differ from the situation

when $L_a \gg L$. However, if we take a comparison of “the constriction” in protein pores with the dTTP molecule occupying the CNT pore here, we might find that the resistance of pore 1 is much larger than pore 2 (i.e. the cross-sectional of pore 1 could be rather small when D_a and D are close). This might be the reason that DNA strand and dTTP molecules afford similar results in these 1.2 nm-diameter CNT pores.

We have added a short discussion in the main text to clarify the use of equation 1. “Although dTTP is a small molecule, its diameter d_{dTTP} is close to the constriction size of MspA and α HL or the diameter of CNT d_{SWCNT} . Therefore, the conductance change ΔG_{pore} caused by dTTP translocation could be estimated mainly by the first term in equation 1.”

Minor technical issues:

How are the AFM samples prepared? What causes the noise pattern of the substrate shown in Figure S4b? Is the noise due to surfactants or other molecules used in the process?

Answer: AFM measurements were performed on mica sheets, which were pre-treated with ultrasonic plasma cleaning before measurement. Therefore, the noise observed in Figure S4b may originate from the results of ultrasonic plasma cleaning. The noise caused by surfactants or other molecules would not be evenly distributed.

It is difficult to see the ultrashort 5-10 nm nanotubes in Figure S5a. Is there a high-resolution AFM image to better show those ultrashort nanotubes? The nanotube length is critical to understand the data reported here.

Answer: Thanks for the point. We have replaced Figure S5a with a higher resolution image which clearly illustrates some ultrashort nanotubes.

The RBM peaks for some of the ultrashort nanotube fractions, such as Figure S6d (Fraction 2 and 4), are unusually sharp. Are they really Raman signals from the nanotube fractions? What is the excitation line?

Answer: We have repeated the Raman micro-spectrometry measurements several times and the results are reproducible. The RBM peaks for ultrashort SWCNTs are indeed sharper than expected. Is it possible that the RBM peaks of SWCNTs become more prominent when its length is decreased to a couple of tens nanometers?

The excitation line is 532 nm.